# Transposon mutagenesis in *Mycobacterium abscessus* identifies an essential penicillin-binding protein involved in septal peptidoglycan synthesis and antibiotic sensitivity

Chidiebere Akusobi[1], Bouchra S Benghomari[2], Junhao Zhu[1], Ian D Wolf[1], Shreya Singhvi[3], Charles L Dulberger[1], Thomas R Ioerger[4], Eric J Rubin[1]*

[1]Department of Immunology and Infectious Diseases, Harvard T.H. Chan School of Public Health, Boston, United States; [2]Department of Biology, Northeastern University, Boston, United States; [3]Department of Molecular and Cell Biology, University of California, Berkeley, Berkeley, United States; [4]Department of Computer Science and Engineering, Texas A&M University, College Station, United States

*For correspondence:
erubin@hsph.harvard.edu

Competing interest: The authors declare that no competing interests exist.

**Abstract** *Mycobacterium abscessus* (*Mab*) is a rapidly growing non-tuberculous mycobacterium (NTM) that causes a wide range of infections. Treatment of *Mab* infections is difficult because the bacterium is intrinsically resistant to many classes of antibiotics. Developing new and effective treatments against *Mab* requires a better understanding of the unique vulnerabilities that can be targeted for future drug development. To achieve this, we identified essential genes in *Mab* by conducting transposon sequencing (TnSeq) on the reference *Mab* strain ATCC 19977. We generated ~51,000 unique transposon mutants and used this high-density library to identify 362 essential genes for in vitro growth. To investigate species-specific vulnerabilities in *Mab*, we further characterized *MAB_3167c*, a predicted penicillin-binding protein and hypothetical lipoprotein (PBP-lipo) that is essential in *Mab* and non-essential in *Mycobacterium tuberculosis* (*Mtb*). We found that PBP-lipo primarily localizes to the subpolar region and later to the septum as cells prepare to divide. Depletion of *Mab* PBP-lipo causes cells to elongate, develop ectopic branches, and form multiple septa. Knockdown of PBP-lipo along with PbpB, DacB1, and a carboxypeptidase, MAB_0519 lead to synergistic growth arrest. In contrast, these genetic interactions were absent in the *Mtb* model organism, *Mycobacterium smegmatis*, indicating that the PBP-lipo homologs in the two species exist in distinct genetic networks. Finally, repressing PBP-lipo sensitized the reference strain and 11 *Mab* clinical isolates to several classes of antibiotics, including the β-lactams, ampicillin, and amoxicillin by greater than 128-fold. Altogether, this study presents PBP-lipo as a key enzyme to study *Mab*-specific processes in cell wall synthesis and importantly positions PBP-lipo as an attractive drug target to treat *Mab* infections.

## Editor's evaluation

This study reports the results of a transposon inactivation screen to identify essential genes in Mycobacterium abscessus. The authors investigate one hit, the gene encoding a putative class B penicillin-binding protein, PBP-lipo. They confirm that the PBP-lipo gene is essential despite the presence of a homologous gene and that PBP-lipo is present in other mycobacteria, but not essential in these. They further characterize the consequences of PBP-lipo gene depletion in M. abscessus

and demonstrate that the gene product is required for maintaining cell morphology, whilst also participating in a network with other cell wall enzymes.

## Introduction

*Mycobacterium abscessus* (*Mab*) is the most common cause of human disease among the rapidly growing non-tuberculous mycobacteria (NTM). It causes a wide range of illnesses including lung, skin, and soft-tissue infections, as well as disseminated disease (*Bryant et al., 2016*; *Primm et al., 2004*; *Johansen et al., 2020*). While the incidence of *Mab* infections is rising worldwide, treating *Mab* infections remains difficult. The bacterium has intrinsic and acquired resistance mechanisms to many classes of antibiotics, including the standard drugs used to treat tuberculosis (*Nessar et al., 2012*; *Pang et al., 2015*). The current treatment regimen for *Mab* requires taking a combination of multiple antibiotics for up to 18 months and is often associated with severe toxicity and routinely ends in treatment failure (*Jarand et al., 2011*). Thus, there is an urgent need for new and effective drugs to treat the emerging global public health threat of *Mab* infections (*Bryant et al., 2021*).

Developing novel antibiotic treatments for *Mab* requires a better understanding of essential biological processes in the bacterium that can be targeted. Identifying essential genes is the first step in this process, and by combining transposon mutagenesis with massive parallel sequencing, transposon sequencing (TnSeq) has been an effective tool in determining gene essentiality on a genome-wide scale in other microbes (*van Opijnen and Camilli, 2013*). However, while many TnSeq screens have been conducted in *Mycobacterium tuberculosis* (*Mtb*) (*DeJesus et al., 2017*), up until recently, only two *Mab* TnSeq screens have been published (*Foreman et al., 2020*; *Laencina et al., 2018*). While both *Mab* TnSeq screens provided tools and important insights into *Mab* biology, neither utilized a mutant library with high enough density to comprehensively identify essential genes. This changed with the recent publication from Rifat et al., who generated robust high-density transposon libraries of *Mab* and identified a comprehensive list of genetic elements essential for in vitro growth (*Rifat et al., 2021*). Identification of these essential genes serves as an excellent starting point to classify attractive targets for drug discovery.

One major target of antibiotics is the mycobacterial cell wall whose biogenesis is essential for bacterial growth (*Wong et al., 2013*; *Bhat et al., 2017*). The structure of the mycobacterial cell wall is unique among bacteria because it is composed of three macromolecules: peptidoglycan (PG), arabinogalactan, and mycolic acids. Together, these layers envelop mycobacteria with a thick, waxy coat that forms a permeability barrier to many drugs (*Jarlier and Nikaido, 1994*; *Nasiri et al., 2017*). The foundational layer is comprised of PG, which consists of glycan sugars cross-linked by short peptides forming a continuous, net-like structure that maintains cell shape and prevents rupture. Mycobacteria build new PG at the cell poles aided by Class A penicillin-binding proteins (PBPs) that perform two reactions: a transglycosylation reaction that polymerizes PG monomers to existing glycan strands and a transpeptidation reaction that cross-links glycan strands to each other (*Kieser and Rubin, 2014*). Class B PBPs solely perform the cross-linking reactions (*Arora et al., 2018*; *Slayden and Belisle, 2009*). A hallmark of PBPs is that they are inhibited by β-lactam antibiotics; however, these antibiotics are typically not used to treat *Mab* infections due to high levels of resistance and the presence of β-lactamases in the genome (*Soroka et al., 2014*).

To identify essential genes for in vitro growth and classify potential new drug targets in *Mab*, we performed TnSeq. We generated high-density transposon libraries of ~51,000 mutants in the reference strain, *M. abscessus* subsp. *abscessus* ATCC 19977 and identified 362 genes required for in vitro growth. We next identified essential genes in *Mab* that were non-essential in *Mtb* in order to study species-specific vulnerabilities in *Mab*. From this list of genes, we further characterized *MAB_3167c*, which encodes a PBP and hypothetical lipoprotein, PBP-lipo, a conserved Class B PBP. We found that PBP-lipo is required for normal PG synthesis and cell division. Knockdown of PBP-lipo sensitizes *Mab* to commonly used β-lactams and other classes of antibiotics. None of these phenotypes were present in the *Mycobacterium smegmatis* (*Msm*) and *Mtb* PBP-lipo knockouts. Thus, we find that PBP-lipo has a unique essential function in *Mab* and is a potential drug target for treating *Mab* infections.

## Results

### Identification of *Mab* ATCC 19977 essential genes for in vitro growth

To identify essential genes in *Mab*, we performed TnSeq on *M. abscessus* subsp. *abscessus* ATCC 19977 (hereafter referred to as *Mab*) using the φMycoMar T7 phage carrying the Himar1 transposon for transduction. This transposon inserts into 'TA' dinucleotide sites and disrupts gene function (*Figure 1A*; *Long et al., 2015*). We transduced three independently grown cultures of *Mab*, yielding triplicate libraries with saturations of 54.2%, 62.3%, and 64.1% for Libraries 1, 2, and 3, respectively (*Supplementary file 1*). The average Pearson $r^2$ correlation of mean read counts for each gene across two libraries was 0.85 indicating good correlation among the libraries (*Figure 1—figure supplement 1*). The transposon insertions across the three libraries were mapped onto the *Mab* genome (*Figure 1B*). Across three libraries, we generated approximately 51,000 unique transposon mutants, as defined by sites of unique transposon insertions across all three libraries. Of note, the *Mab* libraries shared the same non-permissive 'TA' sites for transposon insertion as has been described in other mycobacterial species (*Choudhery et al., 2021*).

The high density of these libraries allowed us to identify essential *Mab* genes using the hidden Markov model (HMM) algorithm implemented in TRANSIT (*DeJesus et al., 2015*). HMM identified 362 genes as essential, 4167 as non-essential, 153 as producing a growth defect when disrupted, and 233 as causing a growth advantage when disrupted (*Figure 1C*, *Supplementary file 2*). As expected, many of the genes identified as essential were involved in key biological processes including protein translation, amino acid metabolism, and cell wall biogenesis based on clusters of orthologous groups analysis (*Figure 1E*).

### Identification and validation of uniquely essential genes in *Mab*

Understanding differences in essential genes between *Mab* and *Mtb* might uncover new ways of specifically targeting *Mab*. Thus, we identified *Mab*-specific essential genes by comparing gene essentiality between mutual orthologs of *Mab* and *Mtb*. We used *Mtb* essentiality data generated from a comprehensive analysis using 14 high-quality TnSeq datasets (*DeJesus et al., 2017*). Of the 2279 mutual orthologs between *Mab* and *Mtb*, 282 genes were essential in both species. We focused on the mutual orthologs that were essential in *Mab*, but non-essential in *Mtb*. These genes could potentially represent *Mab*-specific drug targets. Exactly 20 genes fit into this category, which we refer to as 'uniquely essential' (*Figure 1D*; *Supplementary file 3*).

To understand the quality of predictions by TnSeq, we validated the essentiality of a subset of genes using an anhydrotetracycline (ATc) inducible CRISPRi system developed for use in *Mtb* (*Rock et al., 2017*). Expression of a non-targeting sgRNA control in this system had no impact on cell proliferation (*Figure 2A*). In contrast, repressing canonical essential genes *rpoB*, *gyrB*, and *secY* impaired cell growth in both liquid broth (*Figure 2A*) and solid media (*Figure 2—figure supplement 1A*), indicating that the CRISPRi system developed in *Mtb* functions in *Mab*.

We next validated the essentiality of a subset of the 20 uniquely essential genes in *Mab*: *glnA2*, *icd2*, *sdhA*, and *MAB_4471*. CRISPRi knockdown of these uniquely essential genes impaired growth in liquid media (*Figure 2B*) as well as solid media (*Figure 2—figure supplement 1B*). These results confirm that at least a subset of uniquely essential *Mab* genes identified by TnSeq are required for optimal in vitro growth. The slight rise in $OD_{600}$ later in the time course is consistent with the emergence of escape mutants in the CRISPRi system, which has been previously described (*Peters et al., 2019*; *Caro et al., 2019*).

### PBP-lipo is uniquely essential in *Mab*

One of the uniquely essential genes we identified was *MAB_3167c*, which encodes a hypothesized Class B PBP-lipo. PBP-lipo is predicted to have a single N-terminal transmembrane helix and a transpeptidase domain. We found that *MAB_3167c* lacked transposon insertions across all three replicate libraries (*Figure 3A*). This finding corroborates data from previously generated *Mab* transposon libraries that show lack of insertions in *MAB_3167c* (*Foreman et al., 2020*; *Rifat et al., 2021*). Comparing the essentiality of PBP enzymes in *Mab* with their homologs in *Mtb* revealed that *MAB_3167c* is unique in being the sole PBP whose essentiality differs between the two species. Specifically, PBP-lipo is essential in *Mab* and non-essential in *Mtb* (*Figure 3B*).

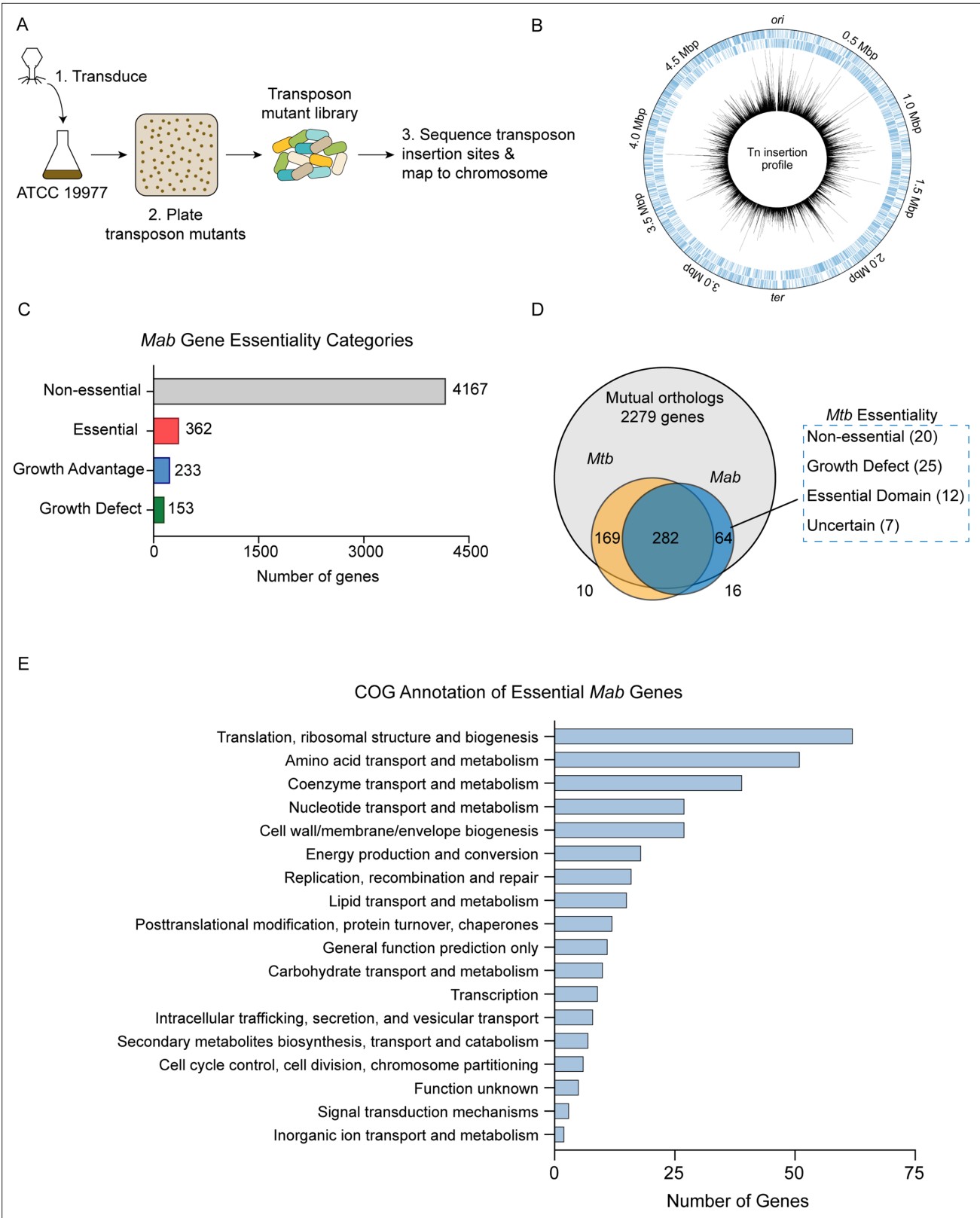

**Figure 1.** Overview of transposon sequencing (TnSeq) analysis. (**A**) Schematic of TnSeq protocol. Phage carrying Himar1 transposon was used to transduce *Mycobacterium abscessus* (*Mab*) subsp. *abscessus* ATCC 19977 cultures. Over 51,000 independent transposon mutants were generated across three libraries. (**B**) Location of transposon insertions in *Mab* genome. Black lines represent the average number of transposon insertions per gene across the three replicates. (**C**) Breakdown of gene essentiality categories as determined by the hidden Markov model (HMM) in TRANSIT. (**D**)

*Figure 1 continued on next page*

*Figure 1 continued*

Essentiality comparison of mutual orthologs between *Mab* and *Mycobacterium tuberculosis* (*Mtb*). (**E**) Clusters of orthologous group (COG) categories of essential genes in *Mab*.

The online version of this article includes the following figure supplement(s) for figure 1:

**Figure supplement 1.** Correlation of *Mycobacterium abscessus* (*Mab*) transposon sequencing (TnSeq) libraries.

To validate that PBP-lipo is required for in vitro growth, we used the CRISPRi system to induce knockdown using constructs encoding 1, 2, or 3 sgRNAs targeting *MAB_3167c* (**Figure 3—figure supplement 1A**). The single, double, and triple sgRNA constructs allowed for the fine tuning of the magnitude of PBP-lipo knockdown, as measured by real-time quantitative PCR (RT-qPCR) (**Figure 3—figure supplement 1B**). We noted that the empty vector control (0 sgRNA) and single sgRNA construct had no detectable effect on the growth of *Mab* (**Figure 3—figure supplement 1C**). In contrast, when *MAB_3167c* was targeted with either 2 or 3 sgRNAs, the growth of *Mab* was inhibited, indicating that PBP-lipo is indeed required for optimal in vitro growth (**Figure 3C**). We also observed inhibition of cell growth on solid media where knockdown of PBP-lipo by 2 sgRNA and 3 sgRNA vectors led to a 1000-fold reduction in colony forming units (CFUs) (**Figure 3—figure supplement 1D**). In contrast to *Mab*, when we deleted the gene encoding PBP-lipo in *Mtb*, *Rv2864c*, the growth of the knockout and wildtype strains was indistinguishable (**Figure 3D**). Similarly, the PBP-lipo homolog in *Msm*, *MSMEG_2584c* when knocked out grew identically to wildtype with no growth defect (**Figure 3—figure supplement 2**).

To complement the PBP-lipo knockdown growth defect phenotype in *Mab*, we recoded a version of PBP-lipo with the protospacer adjacent motif (PAM) and sgRNA-binding sequences mutated while preserving the amino acid sequence. This prevents sgRNAs from binding the exogenous PBP-lipo

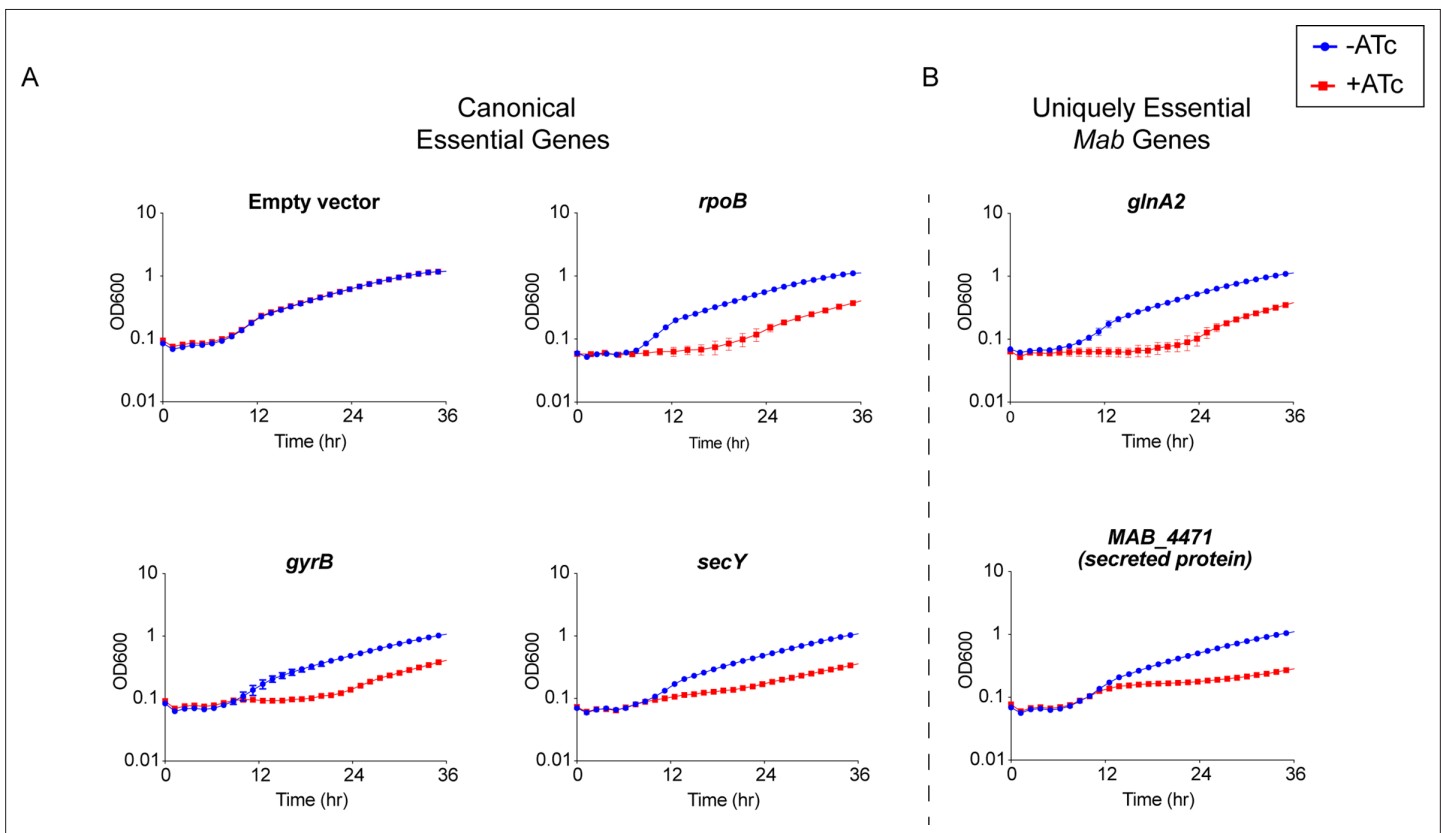

**Figure 2.** Validation of essential genes. (**A**) Growth curve of *Mycobacterium abscessus* (*Mab*) strains with sgRNAs targeting canonical essential genes. (**B**) Growth curve of *Mab* strains with sgRNAs targeting uniquely essential *Mab* genes.

The online version of this article includes the following figure supplement(s) for figure 2:

**Figure supplement 1.** Validation of uniquely essential *Mycobacterium abscessus* (*Mab*) genes.

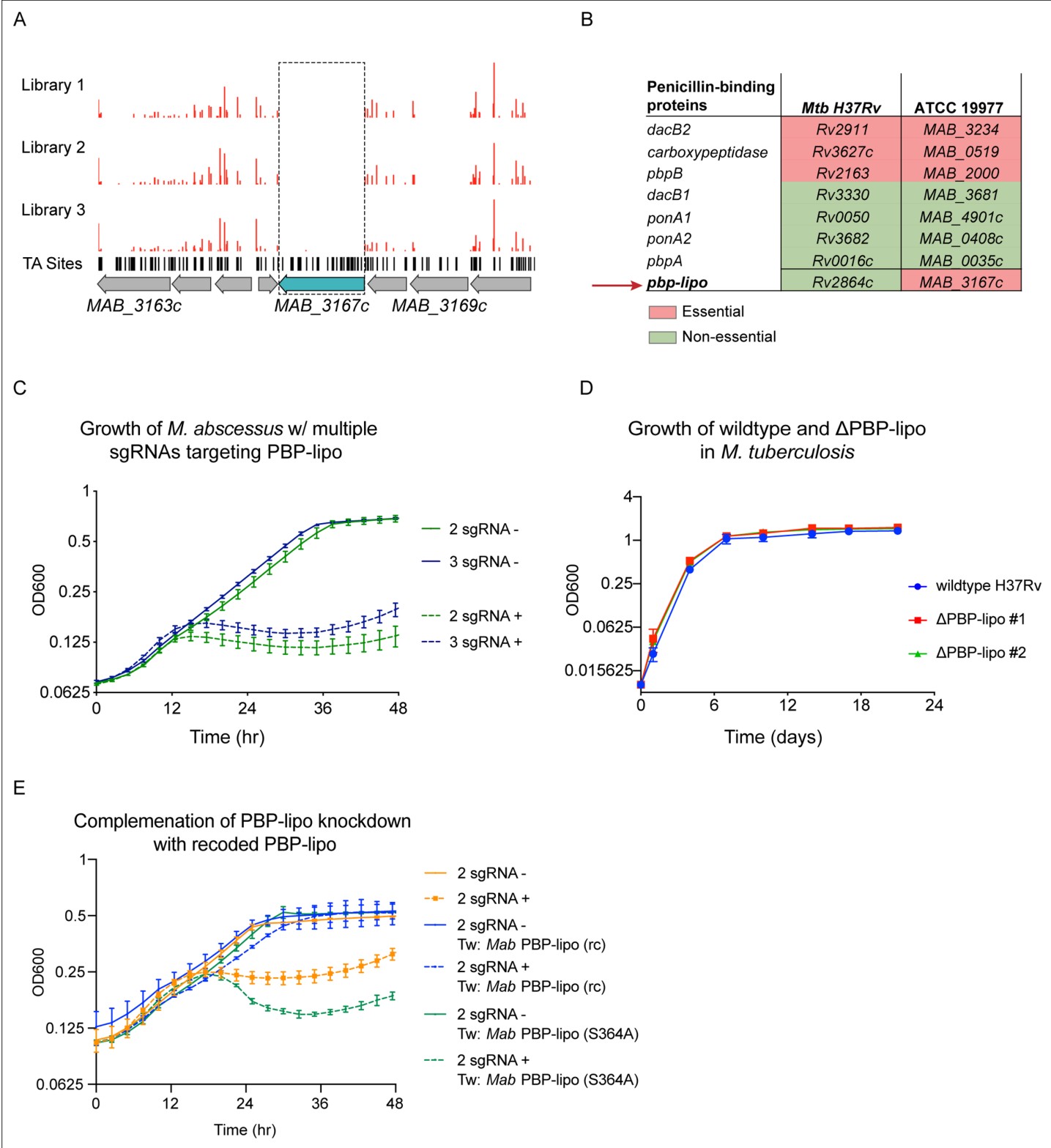

**Figure 3.** Penicillin-binding protein and hypothetical lipoprotein (PBP-lipo) is uniquely essential in *Mycobacterium abscessus* (*Mab*) and non-essential in *Mycobacterium tuberculosis* (*Mtb*). (**A**) Transposon insertion profile of PBP-lipo across three replicate transposon mutant libraries. Red bars indicate the number of transposon insertions. Horizontal black lines indicate 'TA' sites, where the transposon inserts. Genes are schematized with gray arrows, with PBP-lipo colored in blue. Dotted box demarcates lack of insertions in PBP-lipo. (**B**) Comparison of essentiality for PBP orthologs between *Mtb* and *Mycobacterium smegmatis* (*Msm*). (**C**) Growth of *Mab* cultures transformed with CRISPRi plasmid carrying either 2 or 3 sgRNAs targeting PBP-lipo. '-'

*Figure 3 continued*

symbol indicates cultures were grown without anhydrotetracycline (ATc). '+' symbol indicates cultures were grown with anhydrotetracycline (ATc). (**D**) Growth of wildtype *Mtb* and PBP-lipo knockout strains. (**E**) Growth of *Mab* with native PBP-lipo knockdown complemented by recoded (rc) PBP-lipo, which sgRNAs can no longer bind. PBP-lipo (rc) (S364A) is a recoded and catalytically inactive version of the enzyme.

The online version of this article includes the following source data and figure supplement(s) for figure 3:

**Figure supplement 1.** Knockdown of penicillin-binding protein and hypothetical lipoprotein (PBP-lipo) impairs *Mycobacterium abscessus* (*Mab*) growth.

**Figure supplement 2.** Growth of wildtype *Mycobacterium smegmatis* (*Msm*) and penicillin-binding protein and hypothetical lipoprotein (PBP-lipo) knockout strains.

**Figure supplement 3.** *MAB_3167c* encodes a functional penicillin-binding protein (PBP) in *Mycobacterium abscessus* (*Mab*).

**Figure supplement 3—source data 1.** Western blots detecting presence of PBP-lipo-strep.

copy present on the integrated plasmid. When constitutively expressed, this recoded PBP-lipo rescued growth of the native PBP-lipo knockdown (*Figure 3E*). In contrast, a recoded allele that was predicted to be catalytically inactive (S364A) did not rescue growth. All together, this data demonstrates that PBP-lipo's predicted transpeptidase activity is required for *Mab* optimal growth, while PBP-lipo's activity in *Mtb* and *Msm* is non-essential.

## PBP-lipo is required to maintain normal cell elongation and division

A hallmark of PBPs is that they bind β-lactam antibiotics. Therefore, to test whether PBP-lipo is an active PBP in *Mab*, we determined whether PBP-lipo binds bocillin FL, a fluorescent analog of penicillin (*Zhao et al., 1999*). Using a strain that constitutively expresses a strep-tagged PBP-lipo, we identified a fluorescent bocillin FL band at the same molecular weight as PBP-lipo, which was confirmed to be PBP-lipo by Western blot (*Figure 3—figure supplement 3A*). These data suggest that PBP-lipo binds bocillin FL in vivo and is an active PBP. Interestingly, in wildtype *Mab,* we did not detect an obvious PBP-lipo band at the predicted molecular weight of PBP-lipo, 63 kDa, suggesting that native expression levels of PBP-lipo is either low or PBP-lipo is expressed at distinct periods during the cell cycle.

We also tested whether PBP-lipo was indeed a lipoprotein. Based on its amino acid sequence, PBP-lipo has an N-terminal signal sequence that targets the protein for localization to the periplasm. This is followed by a conserved cysteine residue at position 23, which is the canonical site of lipid modification (*Rezwan et al., 2007*). To determine if PBP-lipo is lipidated, we performed a bacterial lipoprotein extraction protocol in *Mab* as described by *Armbruster and Meredith, 2018*. Briefly, cells were lysed and lipoproteins were enriched by TX-114 detergent and later precipitated with acetone. The resulting lipo-protein and non-lipo-protein fractions were then run on an SDS-PAGE gel for further analysis. By Western blot, we detected the presence of PBP-lipo in both the lipoprotein fraction and the non-lipoprotein fraction (*Figure 3—figure supplement 3B*). PBP-lipo being present in the non-lipoprotein fraction is not surprising as this pattern has been observed in early lipoprotein extraction experiments performed in *Mtb* (*Young and Garbe, 1991*). Given these results, we describe PBP-lipo as a hypothetical lipoprotein, but further experiments are needed to confirm if the protein is indeed lipidated.

In mycobacteria, the PG layer is critical for maintaining normal cell shape and altering PBP expression often results in abnormal cell morphology. Thus, we studied the morphology of *Mab* when PBP-lipo was repressed and discovered dramatic morphological changes. *Mab* cells elongate, branch inappropriately, and form ectopic poles of active growth, as evidenced by staining with NADA (3-[(7-nitro-2,1,3-benzoxadiazol-4-yl)amino]-D-alanine), a fluorescent-D-amino acid (FDAA) that binds areas of active PG synthesis (*Kuru et al., 2015*; *Figure 4A*). PBP-lipo knockdown significantly increased the number of cells with ectopic branches (*Figure 4—figure supplement 1A*). Furthermore, during normal growth of *Mab*, a small percentage of cells contain one mid-cell FDAA band, indicating active PG synthesis at the site of septation. However, when PBP-lipo is knocked down, we occasionally find cells with 2 or 3 bands mid-cell, as exemplified in the 2 sgRNA+ panel (*Figure 4A*). PBP-lipo knockdown strains were all significantly longer than their uninduced controls, with the longest average cell length belonging to the strain with the strongest PBP-lipo knockdown (*Figure 4B*). Of note, while the 0 sgRNA+/- cells are morphologically indistinguishable, there is a statistically significant difference in the mean cell lengths, likely due to the effect of ATc on cells (*Figure 4B*). Interestingly, cell width did not change among the strains (*Figure 4—figure supplement 1B*). In comparison, in *Msm* and *Mtb,*

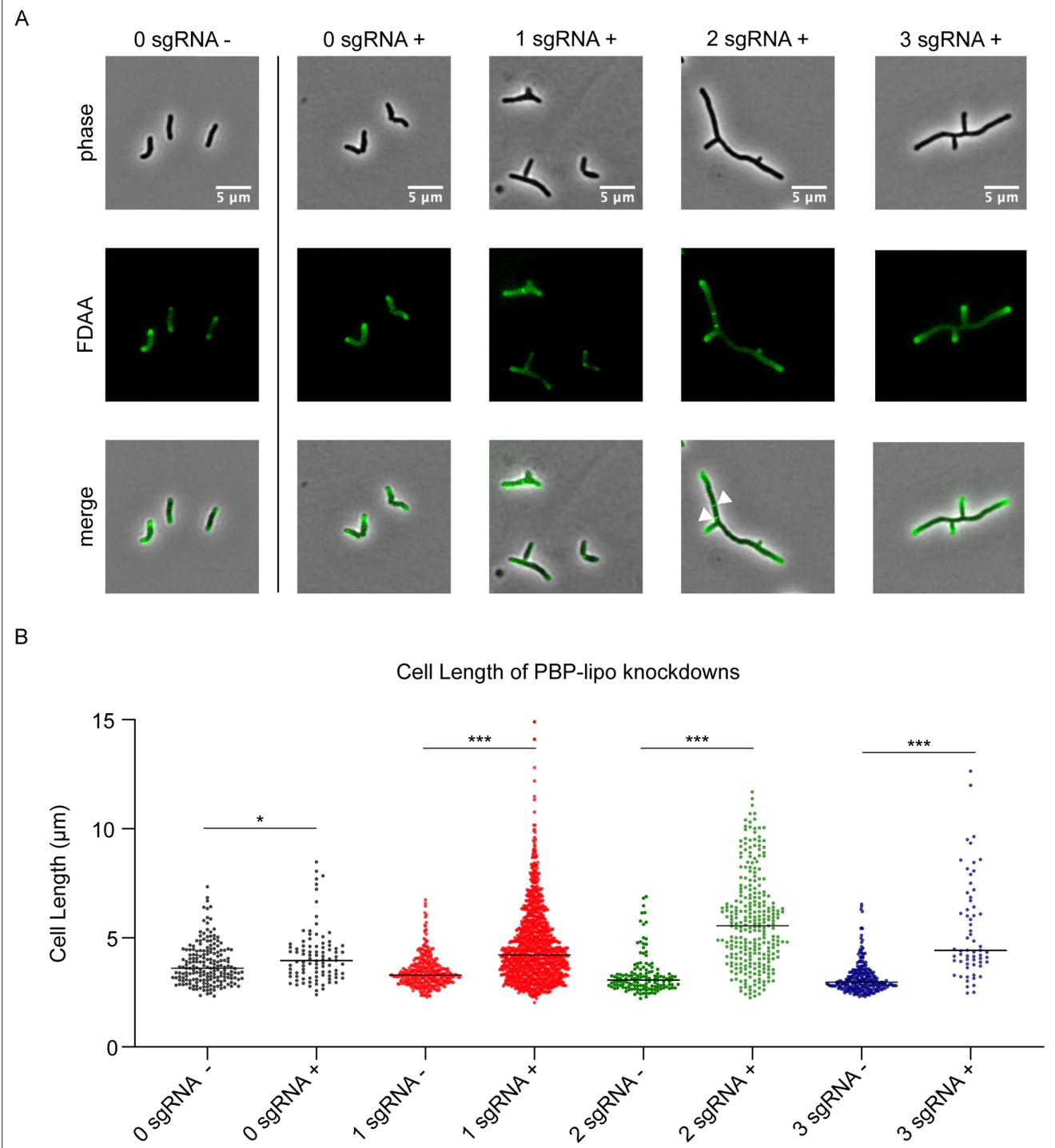

**Figure 4.** Knockdown of penicillin-binding protein and hypothetical lipoprotein (PBP-lipo) disrupts cell morphology. (**A**) Microscopy images of PBP-lipo knockdown cultures. *Mycobacterium abscessus* (*Mab*) strains carrying CRISPRi plasmids with either 0, 1, 2, or 3 sgRNAs targeting PBP-lipo. Arrows indicate sites of multiple septa formation. (**B**) Cell lengths of uninduced and induced strains. Measurements were obtained by GEMATRIA and MOMIA image analysis pipelines (*Zhu et al., 2021*). Student's t test used to calculate the statistical difference in mean cell lengths. ***p<0.0001, *p<0.05.

The online version of this article includes the following figure supplement(s) for figure 4:

**Figure supplement 1.** Morphology of penicillin-binding protein and hypothetical lipoprotein (PBP-lipo) knockdown cells.

**Figure supplement 2.** Knockout of penicillin-binding protein and hypothetical lipoprotein (PBP-lipo) does not alter morphology of *Mycobacterium smegmatis* (*Msm*) and *Mycobacterium tuberculosis* (*Mtb*) cells.

there were no significant differences in morphology and cell length between the wildtype and PBP-lipo knockout strains (*Figure 4—figure supplement 2*). The dramatic changes in *Mab* morphology when PBP-lipo is repressed suggests that proper expression levels of this enzyme are required for maintaining normal elongation and division in *Mab*.

## PBP-lipo localizes to the septum after FtsZ

The morphological defects caused by PBP-lipo knockdown were reminiscent of septal factor depletions observed in *Msm* (*Wu et al., 2018*). Thus, we hypothesized that PBP-lipo localizes to the septum. To test this hypothesis, we N-terminally tagged PBP-lipo with mRFP, which has been previously shown to function in the mycobacterial periplasm (*Baranowski et al., 2018*). We cloned mRFP after the predicted N-terminal signal sequence of PBP-lipo with linkers on the 5' and 3' ends of the fluorescent protein (*Figure 5—figure supplement 1A*). We next validated that this construct produced a full-length fusion protein using Western blot, and detected the correctly sized 91 kDa fusion product (*Figure 5—figure supplement 1B*). Finally, we tested the functionality of the mRFP-PBP-lipo fusion protein by using recoded and non-recoded versions of the construct. The recoded version had a single sgRNA-binding site on PBP-lipo mutated (*Figure 5—figure supplement 1A*). As expected, knockdown of native PBP-lipo lead to growth inhibition while expression of the non-recoded fusion protein partially complemented growth. Reassuringly, the recoded mRFP-PBP-lipo fully complemented the growth defect from the native PBP-lipo knockdown indicating that the fusion protein is indeed functional (*Figure 5—figure supplement 1C*).

Given the morphological defects in the PBP-lipo knockdown, we hypothesized that PBP-lipo localized to the septum. To test this, we imaged PBP-lipo simultaneously with FtsZ, which forms the Z-ring at the divisome. FtsZ is the first protein to localize to the septum and recruits other members of the divisome complex to aid in the formation of the septum and eventual division of the cell (*Adams and Errington, 2009*). To visualize FtsZ, we C-terminally tagged it with mNeonGreen (mNG) and expressed the fusion protein using a weak promoter as higher levels of expression are toxic. After introducing both constructs into wildtype *Mab*, we visualized the fusion proteins, which were expressed in the presence of wildtype PBP-lipo. Using fluorescence microscopy, we discovered that both FtsZ and PBP-lipo localized to the mid-cell, likely in the growing septum. We also identified cells with septal co-localization of both PBP-lipo and FtsZ (*Figure 5A*).

The resulting images were analyzed using the recently published MOMIA and GEMATRIA programs, which quantify the fluorescent signal across cells normalized by cell length (*Zhu et al., 2021*). We first analyzed cells that expressed solely mRFP-PBP-lipo and demonstrated that mid-cell localization of PBP-lipo was only present in the longest cells in the population (*Figure 5B*). These data demonstrate that as cells elongate, PBP-lipo localizes to the septum and is therefore likely involved in septum formation and cell division. We next applied this analysis to cells that co-expressed mRFP-PBP-lipo and FtsZ-mNeonGreen. We found that FtsZ localizes to the septum earlier in the cell life cycle as demonstrated by the higher septal fluorescence signal intensity in shorter cells (*Figure 5C*). In contrast, PBP-lipo localizes to the septum when cells are longer and after FtsZ has already localized to the septum (*Figure 5C*). This sequential recruitment of divisome proteins is well described in several bacteria, including *Msm* (*Wu et al., 2018*). Localization of FtsZ determines the location of the septum, followed by recruitment of PBPs involved in septal PG synthesis and finally enzymes that aid in daughter cell separation (*Kieser and Rubin, 2014*). Our data shows the temporal dynamics of FtsZ and PBP-lipo septal localization as well as highlights cells where septal co-localization of both proteins can be appreciated.

## Knockdown of PBP-lipo disrupts FtsZ regulation and localization

PBP-lipo repression leads to elongated and branched cells with multiple sites of septation as demonstrated by FDAA staining (*Figure 4A*). This indicates that PBP-lipo repression disrupts the regulation of cell division in *Mab*. We hypothesized that this could be due to dysregulated formation of the FtsZ rings responsible for initiating division. To test this hypothesis, we visualized FtsZ in the context of native PBP-lipo knockdown. We used an N-terminal GFP-FtsZ fusion protein expressed off the FtsZ natural promoter to best capture the natural dynamics of expression. Upon PBP-lipo knockdown, we identified cells that formed multiple FtsZ puncta in contrast to the single puncta found in normally growing cells (*Figure 5D*). These multiple FtsZ puncta were distributed across the length of

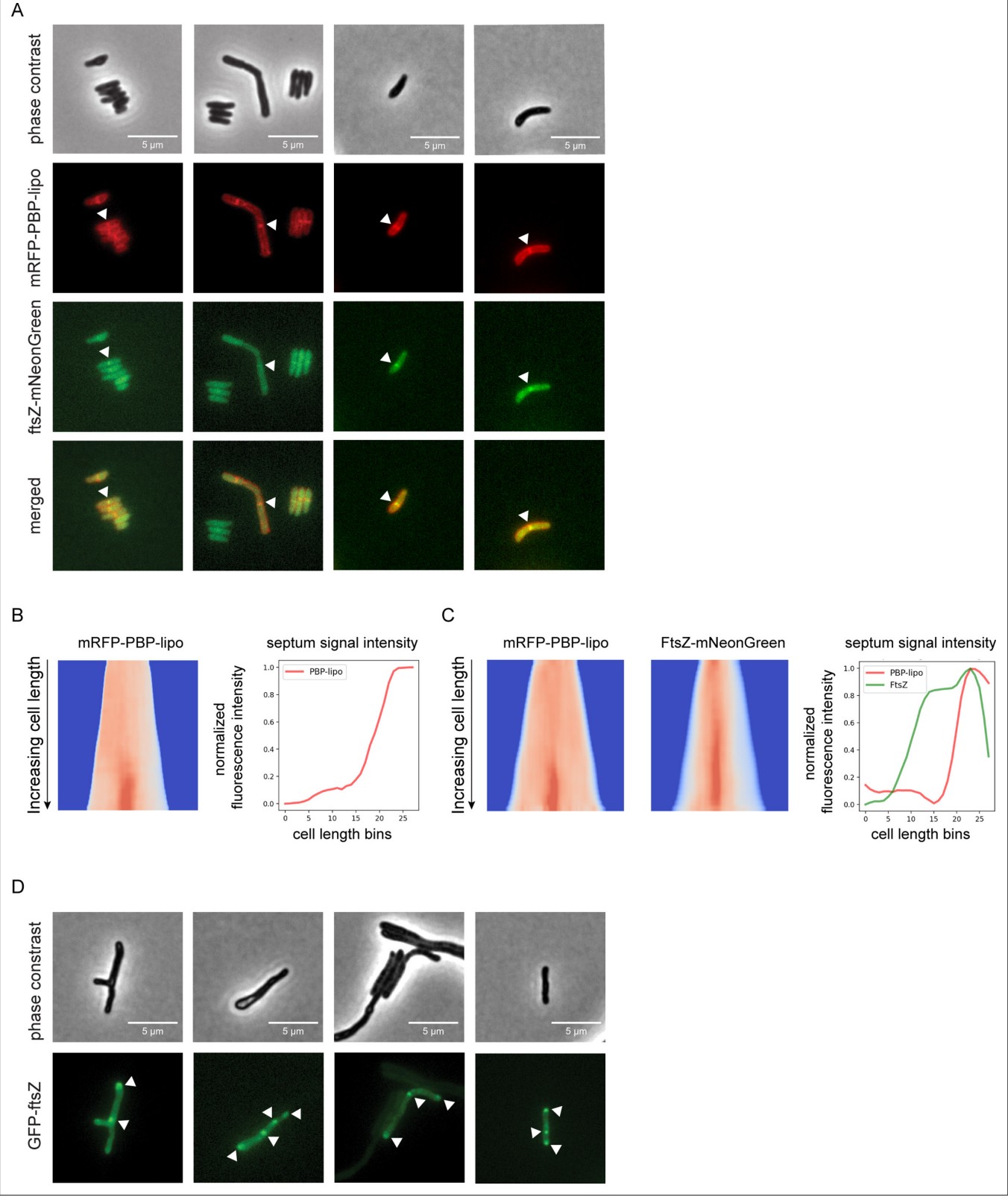

**Figure 5.** Knockdown of penicillin-binding protein and hypothetical lipoprotein (PBP-lipo) disrupts formation of FtsZ rings. (**A**) Microscopy images of N-terminally tagged mRFP-PBP-lipo and C-terminally tagged FtsZ-mNeonGreen. 'Merged' images show overlay of red and green channels. (**B**) (Left) Demograph of mRFP-PBP-lipo. (Right) Quantification of septal fluorescence signal across increasing cell lengths. (**C**) (Left) Demograph of mRFP-PBP-lipo

*Figure 5 continued on next page*

*Figure 5 continued*

and FtsZ-mNeonGreen. (Right) Fluorescence signal arranged by increasing cell length. (**D**) Images of GFP-FtsZ expressed from its natural promoter in the setting of PBP-lipo knockdown.

The online version of this article includes the following source data and figure supplement(s) for figure 5:

**Figure supplement 1.** mRFP-PBP-lipo is a functional fusion protein.

**Figure supplement 1—source data 1.** Western blot detecting presence of mRFP-PBP-lipo-strep.

the cell indicating that repression of PBP-lipo disrupts FtsZ localization. Furthermore, the formation of multiple FtsZ rings likely leads to the creation of multiple septa and eventually the ectopic branching events observed when PBP-lipo is repressed.

## PBP-lipo is incorporated into a unique genetic PG network in *Mab* that does not exist in *Msm*

Given that *Mtb* and *Msm* do not have duplicates of PBP-lipo in their genomes, we hypothesized that there is a functional difference between PBP-lipo in *Mab* and its homologs in *Msm* and *Mtb*. This possibility intrigued us because an alignment of PBP-lipo from various mycobacteria revealed that PBP-lipo$_{Mab}$ has five additional amino acids present only in the *Mab* homolog (*Figure 6—figure supplement 1A*). When in silico structures of PBP-lipo$_{Mab}$ and PBP-lipo$_{Mtb}$ are modeled, these five extra residues form an appendage we hypothesized could impart PBP-lipo$_{Mab}$ with additional ability to bind proteins (*Figure 6—figure supplement 1B*).

To test this hypothesis, we constitutively expressed PBP-lipo$_{Msm}$ in the 2 sgRNA *Mab* strain (hereafter referred to as the *Msm* complement strain). As expected, knockdown of the native *Mab* PBP-lipo impairs growth. However, PBP-lipo$_{Msm}$ expression in the 2 sgRNA strain rescued *Mab* growth (*Figure 6A*). Furthermore, expressing PBP-lipo$_{Msm}$ eliminated the branching phenotype of the native PBP-lipo knockdown and partially reversed the elongated phenotype as well (*Figure 6B*). The recoded version of PBP-lipo$_{Mab}$ also reversed the branching of cells and partially complemented the elongated phenotype (*Figure 6B*). In contrast, the catalytically inactive version of PBP-lipo$_{Mab}$-(S364A) did not complement, with cells remaining elongated and branched (*Figure 6B*, *Figure 6—figure supplement 2*). These results demonstrate that PBP-lipo$_{Msm}$ can rescue knockdown of the endogenous *Mab* enzyme. From these data, we conclude that the essentiality of PBP-lipo$_{Mab}$ is not solely due to a unique functional property that is present only in the *Mab* homolog.

An alternative hypothesis is PBP-lipo participates in a PBP interaction network in *Mab* that is distinct from respective PBP networks in *Msm* and *Mtb*. In short, while PBP-lipo is required in the *Mab* PBP network, its function is likely redundant in the networks of *Msm* and *Mtb*. This notion of functionally distinct PBP networks has been previously described in mycobacteria (*Kieser et al., 2015a*). To interrogate the PBP network of *Mab*, we tested which PBPs genetically interacted with PBP-lipo. To perform this experiment, we cloned CRISPRi constructs with sgRNAs that would weakly knock down each PBP in *Mab* (PBPx), so that growth was not significantly impaired. We also cloned combinatorial CRISPRi constructs where PBP-lipo and another PBP were weakly repressed. Using this dual sgRNA vector, we could simultaneously knock down PBP-lipo and a PBP of interest.

Using combinatorial knockdowns of each PBP, we compared growth inhibition when a single PBP was knocked down versus when PBP-lipo and a second PBP were jointly repressed. Of the eight combinations we tested, we found that knockdown of three PBPs with PBP-lipo lead to synergistic growth arrest. These genes were *pbpB*, *dacB1*, and *MAB_0519*, which encodes a protein with a hypothesized carboxypeptidase domain (*Figure 6C*). This genetic synergy was also observed in liquid culture for *pbpB* and *dacB1* (*Figure 6—figure supplement 3A*). The homolog of PbpB is a well-studied enzyme in both *Msm*, *Mtb*, and the model organisms *Escherichia coli* and *Bacillus subtilis*, where it is known as FtsI. In all these organisms, PbpB localizes to the septum and is largely responsible for septal PG synthesis (*Slayden and Belisle, 2009*; *Wissel and Weiss, 2004*; *Datta et al., 2006*). Interestingly, overexpression of PbpB rescued the growth defect caused by knockdown of PBP-lipo (*Figure 6—figure supplement 3B*). Similarly, exogenous expression of PbpB fully reversed the morphological defects of PBP-lipo repression (*Figure 6—figure supplement 3C*). These data indicate that PBP-lipo and PbpB have overlapping functions at the septum in *Mab*. However, because both enzymes are individually essential, PbpB and PBP-lipo must have distinct functions as well.

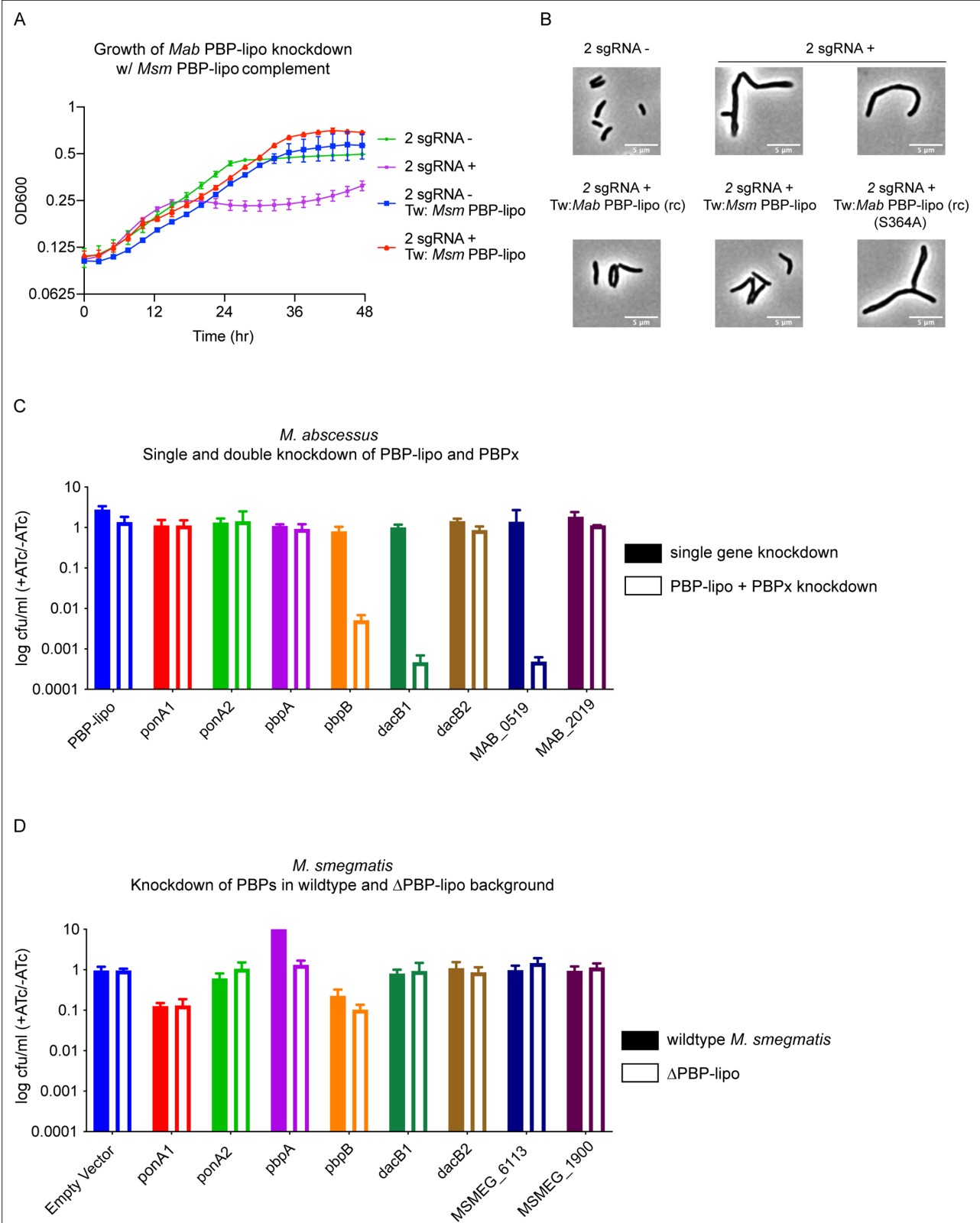

**Figure 6.** Penicillin-binding protein and hypothetical lipoprotein (PBP-lipo) network in *Mycobacterium abscessus (Mab)* does not exist in *Mycobacterium smegmatis (Msm)*. (**A**) Growth curve of 2 sgRNA strain targeting PBP-lipo and 2 sgRNA strain constitutively expressing PBP-lipo$_{Msm}$. (**B**) Microscopy images of 2 sgRNA and PBP-lipo complement strains. (**C**) Genetic synergy of PBP-lipo with other PBPs in *Mab*. Solid bar represents colony forming unit (CFU) of strains where a single PBP was knocked down. Open bar represents strains where PBP-lipo was knocked down in combination with the listed

*Figure 6 continued on next page*

*Figure 6 continued*

PBP. (**D**) Genetic synergy of PBPs and PBP-lipo in *Msm*. CRISPRi plasmids carrying sgRNAs targeting each of the PBPs in *Msm* were transformed into wildtype *Msm* (closed bar) and the ΔPBP-lipo mutant (open bar).

The online version of this article includes the following source data and figure supplement(s) for figure 6:

**Figure supplement 1.** Structural comparison of penicillin-binding protein and hypothetical lipoprotein (PBP-lipo) across mycobacteria.

**Figure supplement 2.** Cell length of penicillin-binding protein and hypothetical lipoprotein (PBP-lipo) knockdown and complement strains.

**Figure supplement 3.** Genetic interactions of penicillin-binding protein and hypothetical lipoprotein (PBP-lipo) and PBPs.

**Figure supplement 4.** DacB1 localizes to the septum of *Mycobacterium abscessus* (*Mab*) and co-localizes with penicillin-binding protein and hypothetical lipoprotein (PBP-lipo).

**Figure supplement 4—source data 1.** Western blot detecting presence of DacB1-GFPmut3.

**Figure supplement 5.** Genetic interactions of penicillin-binding proteins (PBPs) in *Mycobacterium smegmatis* (*Msm*) and *Mycobacterium abscessus* (*Mab*).

**Figure supplement 6.** Knockdown of penicillin-binding protein and hypothetical lipoprotein (PBP-lipo) increases rate of calcein accumulation in *Mycobacterium abscessus* (*Mab*).

**Figure supplement 7.** Knockdown of penicillin-binding protein and hypothetical lipoprotein (PBP-lipo) impairs growth in clinical isolates.

To test the importance of the genetic interaction of DacB1 with PBP-lipo, we created a DacB1-GFP fusion, which we validated as being fully intact by Western blot (*Figure 6—figure supplement 4A*). Interestingly, like PbpB and PBP-lipo, DacB1 localizes to the septum (*Figure 6—figure supplement 4B*). We also observed co-localization of DacB1 and PBP-lipo at the septal region of the cell (*Figure 6—figure supplement 4C*). Using the MOMIA and GEMATRIA programs, we quantified the fluorescent signal from DacB1 and PBP-lipo at the septum and binned the cells by cell length. These analyses demonstrate that both DacB1 and PBP-lipo localize to the septum as cells elongate, with DacB1 arriving slightly before PBP-lipo to the septum (*Figure 6—figure supplement 4D*). These data suggest that PBP-lipo genetically interacts with PbpB and DacB1, and that all three enzymes localize to the same region of the cell and could potentially coordinate septal PG synthesis in a complex together.

The difference in essentiality of PBP-lipo in different species could be due to altered genetic interactions. To test this in *Msm,* we used CRISPRi to knock down *Msm* PBP homologues in the wildtype *Msm* and the PBP-lipo knockout background. In this experiment, genes that interacted with PBP-lipo would show impaired cell growth when repressed in the PBP-lipo knockout. Interestingly, we did not observe genetic interactions between any of the PBPs tested and PBP-lipo (*Figure 6D*). Importantly, the level of knockdown for genes that showed synergy in *Mab* was comparable to the level of knockdown in the *Msm* PBP-lipo knockout, as measured by qPCR (*Figure 6—figure supplement 5A*). We also did not detect synergy in additional PBPs present in the *Msm* genome with no known homologs in *Mab* (*Figure 6—figure supplement 5B*).

These results might suggest that *Msm* PBP-lipo might have different interacting partners. To test this, we generated high-density transposon mutant libraries in both the wildtype and PBP-lipo knockout strains of *Msm* to identify genes that were conditionally essential in the knockout background. The saturation of the two wildtype libraries were 51.9% and 53.4%, while the saturation of the PBP-lipo knockout library was 62.4%. Using the resampling method in TRANSIT, our TnSeq data revealed that there were no conditionally essential genes in the ΔPBP-lipo *Msm* background (*Supplementary file 4*). These data strongly suggest that PBP-lipo in *Msm* exists in a robust genetic network with multiple functional redundancies. To uncover PBP-lipo's function in *Msm*, it is possible that double and triple knockout mutants of PG synthesis enzymes would need to be generated.

## Repressing PBP-lipo sensitizes *Mab* to a wide range of antibiotics

Given PBP-lipo's essentiality in *Mab* and the genetic interactions it has with other PBPs, we hypothesized that repressing PBP-lipo would affect *Mab*'s sensitivity to antibiotics, including β-lactams. To test this hypothesis, we measured the sensitivity of PBP-lipo knockdown cells to 21 antibiotics that target translation, transcription, and DNA replication in both *Mab* and *Msm*.

Wildtype *Mab* is typically resistant to β-lactams amoxicillin and ampicillin with minimum inhibitory concentrations (MICs) greater than >256 μg/ml. When PBP-lipo is knocked down, the MIC for both antibiotics dropped to 2 μg/ml, indicating a greater than 128-fold sensitization (*Table 1*). These results

**Table 1.** Minimum inhibitory concentration (MIC) (μg/ml) of ATCC 19977 +/-penicillin-binding protein and hypothetical lipoprotein (PBP-lipo) knockdown.

| Antibiotic | -ATc | +ATc | Fold difference in MIC (-ATc/+ATc) |
|---|---|---|---|
| **Cell wall** | | | |
| Ampicillin | >256 | 2 | >128× |
| Amoxicillin | >256 | 2 | >128× |
| Faropenem | 8 | 1 | 8× |
| Cefoxitin | 8 | 2 | 4× |
| Vancomycin | 8 | 2 | 4× |
| Imipenem | >64 | >64 | - |
| Ticarcillin | >64 | >64 | - |
| Ceftazidime | >64 | >64 | - |
| Cephalexin | >64 | >64 | - |
| Ceftazidime | >64 | >64 | - |
| D-Cycloserine | >64 | >64 | - |
| Ethambutol | >64 | >64 | - |
| Isoniazid | >512 | >512 | - |
| | | | |
| **Ribosome** | | | |
| Clarithromycin | 16 | 0.125 | 128× |
| Erythromycin | 32 | 0.5 | 64× |
| Amikacin | 16 | 4 | 4× |
| Clindamycin | >64 | 64 | - |
| | | | |
| **RNA polymerase** | | | |
| Rifampicin | 32 | 2 | 16× |
| | | | |
| **DNA gyrase** | | | |
| Ofloxacin | 8 | 2 | 4× |
| | | | |
| **Other** | | | |
| Pyrazinamide | >64 | >64 | - |
| Pretomanid | >64 | >64 | - |

indicate a strong chemical-genetic interaction between PBP-lipo and both ampicillin and amoxicillin. Interestingly, not all β-lactams had shifts in their MIC when PBP-lipo was knocked down. This suggests that there is a specific chemical-genetic interaction between PBP-lipo or its partner PBPs and ampicillin and amoxicillin. MIC changes for other β-lactams were less drastic, with a 4- to 8-fold reduction observed for faropenem and cefoxitin (**Table 1**). PBP-lipo knockdown also affects susceptibility to antibiotics that do not target the cell wall. The macrolides, clarithromycin and erythromycin, were 128-fold and 64-fold more potent respectively in the PBP-lipo knockdown cells. A 16-fold reduction in MIC was also observed for RNA polymerase inhibitor, rifampicin.

The MIC differences in rifampicin and clarithromycin are notable because these antibiotics do not target the cell wall. Thus, the mechanism of synergy between PBP-lipo repression and these antibiotics likely differs from the synergy with ampicillin and amoxicillin. We hypothesized that PBP-lipo repression might increase the permeability of the cell wall, allowing more drug to enter the cell. To test this hypothesis, we measured the accumulation of calcein into *Mab*, which is one proxy for cell permeability. Calcein accumulated more rapidly in PBP-lipo knockdown cells, indicating that PBP-lipo increased the permeability of *Mab* cells (*Figure 6—figure supplement 6*). This increased permeability might lead to increased antibiotic access to the cell, thus sensitizing the bacteria to the antibiotics.

When the MICs of the same 21 antibiotics tested in *Mab* were measured in the PBP-lipo knockout in *Msm,* we observed no difference in the MICs (*Supplementary file 5*). We also measured the MICs of a subset of these antibiotics in the PBP-lipo knockout in *Mtb* and found no difference in MICs between the wildtype and knockout strain (*Supplementary file 6*). These data suggest that PBP-lipo's unique function in *Mab's* PG synthesis network likely contributes to the species' sensitivity to antibiotics when the enzyme is repressed.

**Table 3.** Minimum inhibitory concentration (MIC) (μg/ml) of *Mycobacterium abscessus* (*Mab*) clinical isolates +/-penicillin-binding protein and hypothetical lipoprotein (PBP-lipo) knockdown (rifampicin and clarithromycin).

| Clinical isolate | Rifampicin | | Clarithromycin | |
|---|---|---|---|---|
| | -ATc | +ATc | -ATc | +ATc |
| ATCC 19977 | 32 | 2 | 16 | 0.125 |
| T35 | 4 | 0.25 | 2 | 0.25 |
| T37 | 16 | 1 | 2 | 0.25 |
| T49 | 64 | 4 | 2 | 0.25 |
| T50 | 32 | 1 | 16 | 0.25 |
| T51 | 64 | 4 | 16 | 0.5 |
| T53 | 32 | 1 | 8 | 0.5 |
| T56 | 8 | 1 | 2 | 0.25 |
| BWH-B | 32 | 1 | 16 | 0.25 |
| BWH-C | 32 | 2 | 32 | 1 |
| BWH-E | 16 | 1 | 16 | 0.125 |

## Repression of PBP-lipo sensitizes *Mab* clinical isolates to antibiotics

The antibiotic experiments were conducted on the lab-adapted ATCC 19977 reference strain on *Mab*, after being isolated from a knee abscess in the 1950s and propagated in the lab since (*Johansen et al., 2020*; *Tortoli et al., 2016*). Work performed in *Mtb* has demonstrated that clinical isolates and lab-adapted strains respond to antibiotics stressors differently (*Carey et al., 2018*). As a result, we tested whether knockdown of PBP-lipo in *Mab* clinical isolates led to the same changes in antibiotic sensitivity as observed in the reference strain.

We found that knockdown of PBP-lipo using the 3 sgRNA construct impaired cell growth in all 11 clinical isolates leading to a 100- to 10,000-fold difference in CFUs (*Figure 6—figure supplement 7A*). We also confirmed that in a subset of the clinical isolates PBP-lipo knockdown lead to elongated and inappropriately branched cells, as observed in the reference strain (*Figure 6—figure supplement 7B*). After confirming the PBP-lipo phenotypes in the clinical isolates, we measured the MICs of ampicillin, amoxicillin, rifampicin, and clarithromycin on PBP-lipo knockdowns of the clinical isolates.

For amoxicillin and ampicillin, PBP-lipo knockdown reduced the MICs by >32- to>512-fold (*Table 2*). Finally, we observed an 8- to 32×-fold

**Table 2.** Minimum inhibitory concentration (MIC) (μg/ml) of *Mycobacterium abscessus* (*Mab*) clinical isolates +/-penicillin-binding protein and hypothetical lipoprotein (PBP-lipo) knockdown (cell wall antibiotics).

| Clinical isolate | Ampicillin | | Amoxicillin | |
|---|---|---|---|---|
| | -ATc | +ATc | -ATc | +ATc |
| ATCC 19977 | >256 | 2 | >256 | 2 |
| T35 | >256 | <0.5 | >256 | <0.5 |
| T37 | >256 | <0.5 | >256 | <0.5 |
| T49 | >256 | 4 | >256 | 4 |
| T50 | >256 | 4 | >256 | 4 |
| T51 | >256 | 2 | >256 | 2 |
| T53 | >256 | 4 | >256 | 2 |
| T56 | >256 | <0.5 | >256 | <0.5 |
| BWH-B | >256 | 8 | >256 | 2 |
| BWH-C | >256 | 4 | >256 | 8 |
| BWH-E | >256 | 4 | >256 | 2 |

difference in the MICs for rifampicin and 8- to 128×-fold difference in the MICs for clarithromycin (*Table 3*). Altogether, the antibiotic sensitivity data demonstrate that the phenotypic consequences of repressing PBP-lipo is not just limited to the lab-adapted reference strain, but is observed in *Mab* clinical isolates as well.

## Discussion

Developing new drugs and drug regimens against *Mab* requires a better understanding of the bacterium's essential processes. This includes identifying essential *Mab* genes for further characterization as potential drug targets. To do this, we generated high-density transposon libraries to identify essential genes in the reference strain, *M. abscessus* subsp. *abscessus* ATCC 19977. Importantly, *Mab* transposon libraries with fewer than 10,000 unique mutants have been previously published, but these studies did not identify the full list of essential genes in *Mab* (*Foreman et al., 2020*; *Laencina et al., 2018*). Recently, Rifat et al. successfully generated high-density libraries and were the first group to identify essential genes and genomic elements in the *Mab* genome (*Rifat et al., 2021*). All together, they identified 326 essential genes and compared this list to essential genes identified in *Mtb* and *Mycobacterium avium*. Similar to our study, Rifat et al. highlighted uniquely essential genes in *Mab*, which they also state deserve further study as these genes could represent *Mab*-specific drug targets (*Rifat et al., 2021*).

In addition to identifying essential genes in *Mab*, our study builds upon Rifat et al.'s work by using CRISPRi to repress gene expression and validate predicted essential genes as required for in vitro growth (*Rock et al., 2017*). Furthermore, we found that by targeting genes with multiple sgRNAs, CRISPRi can achieve significant gene repression, allowing for the investigation of essential genes and providing an alternative to gene knockouts. Constructing gene knockouts in *Mab* can be inefficient (*Medjahed and Reyrat, 2009*); thus, CRISPRi provides a convenient tool for genetic manipulation in *Mab*. Overall, the newly generated *Mab* transposon libraries can be used to determine genes required for growth in a variety of conditions including infection models, biofilms, and antibiotic stress.

Both Rifat et al.'s and our TnSeq data identified PBP-lipo as the sole PBP that is essential in *Mab* and non-essential in *Mtb* (*Rifat et al., 2021*). Using CRISPRi, we demonstrated the necessity of PBP-lipo expression for *Mab* to maintain normal morphology and to divide properly. When PBP-lipo is repressed, cells elongate, branch, and form ectopic poles. We localized PBP-lipo to the subpolar regions of the cell, and as cells elongate and prepare for division, PBP-lipo localizes to the septum. Furthermore, depleting PBP-lipo led to the creation of multiple Z-rings in the cell and likely the aberrant start of cell division and formation of branched cells. Overall, the constellation of PBP-lipo knockdown phenotypes mimics septal-factor depletions observed in *Msm*, which also cause cells to elongate, branch, and form ectopic poles (*Wu et al., 2018*). These data suggest that PBP-lipo is involved with PG synthesis at the septum and that depletion of the enzyme leads to dysregulated septal PG synthesis and cell division.

Given PBP-lipo's localization and effect on cell division, we propose that PBP-lipo acts as an essential septal PG synthesis enzyme in *Mab*. In *Mtb* and *Msm*, this is accomplished by the enzyme *pbpB*, which is essential for growth. Additionally, when *pbpB* is repressed in these species, cells appear morphologically similar to the PBP-lipo knockdown in *Mab* (*Wu et al., 2018*; *Slayden and Belisle, 2009*). Interestingly, PBP-lipo knockouts in both *Msm* and *Mtb* are morphologically identical to wild-type cells (*Figure 4—figure supplement 2*). These data underscore that biological insights from *Msm* and *Mtb* should not be de facto applied to *Mab*. We hypothesize that PBP-lipo is a member of the *Mab* divisome complex, a collection of structural proteins and enzymes that coordinates cell division (*Kieser and Rubin, 2014*). This hypothesis was supported by the co-localization of PBP-lipo with FtsZ, a key initiating component of the bacterial divisome (*Adams and Errington, 2009*). Future studies are needed to determine if PBP-lipo interacts with known septal factors, such as FtsQ and FtsW, whose functions in the divisome have been described in *Msm* (*Datta et al., 2006*; *Rajagopalan et al., 2005*; *Datta et al., 2002*). Future work may also reveal novel PBP-lipo-binding factors that coordinate PG synthesis and cell division in *Mab* specifically.

Phenotypes characteristic of the PBP-lipo knockdown in *Mab* were absent in the *Mtb* and *Msm* PBP-lipo knockouts. Both *Mtb* and *Msm* do not have a duplicate copy of PBP-lipo in their genomes. Thus, to explain the differential essentiality of PBP-lipo across these mycobacterial species, one hypothesis we explored was differences in the genetic networks of PG synthesis enzymes. Construction of the

PG layer is a complex process that involves a collection of enzymes that work in concert to ensure proper synthesis and remodeling. Previous work has shown that PG synthesis enzymes belong to genetic and spatiotemporal networks that do not necessarily replicate across closely related species (*Kieser et al., 2015a*; *Botella et al., 2017*). Thus, while the synthetic enzymes to construct the PG layer may be homologous between species, how the enzymes are networked and subsequently function together may differ. We demonstrated that PBP-lipo genetically interacts with *pbpB*, *dacB1*, and *MAB_0519*. In contrast, TnSeq on the *Msm Δpbp-lipo* strain did not reveal interactions with any other gene in the genome. This demonstrates that despite being homologs, PBP-lipo in *Mab* and *Msm* are incorporated into different PG synthesis networks. In *Mab*, this network renders PBP-lipo's function as essential for growth and proper cell division, whereas in *Msm*, PBP-lipo is dispensable. Interestingly, in *Mtb* PBP-lipo is required for normal growth in a *ponA2* knockout suggesting that PBP-lipo and PonA2 genetically interact *Mtb* (*Kieser et al., 2015a*) – a genetic interaction that was not observed in *Mab*. This observation supports the hypothesis that PBP-lipo is incorporated in different PG synthesis in *Mtb* as well.

While this work demonstrates the essential function of PBP-lipo for growth and division in *Mab* and not in *Msm* or *Mtb*, all the experiments were performed in standard growth conditions. Studies in other bacteria have demonstrated that homologous PBPs that are otherwise non-essential become required under specific growth conditions. For example, in *Salmonella*, the Class B homolog of PBP3, PBP3-SAL, was required for cell division in acidified intraphagosomal environments (*Castanheira et al., 2017*). Similarly, in *E. coli*, PBP6, which encodes a carboxypeptidase, was more active at lower pH values compared to five other homologous carboxypeptidases (*Peters et al., 2016*). These studies demonstrate that some PBPs become more active and their functions more required in specific growth conditions. Future studies could test PBP-lipo's function under various stress conditions. These experiments may elucidate a condition-specific role for PBP-lipo in *Msm* and *Mtb* that differs from its central function in *Mab*.

Not only is PBP-lipo essential in *Mab*, it also genetically interacts with three other PBPs, *pbpB*, *dacB1*, and *MAB_0519*, which are all hypothesized to be septal associated. PbpB is a D,D-transpeptidase that performs 4,3 cross-linking reactions at the septum. In *Msm*, depleting PbpB leads to cell filamentation and branching reminiscent of the *Mab* PBP-lipo knockdown phenotypes (*Wu et al., 2018*). A potential explanation of this genetic interaction is both PbpB and PBP-lipo perform 4,3-cross-linking at the *Mab* septum. In this model, repressing both enzymes would dramatically reduce 4,3-cross-linking at the septum and lead to cell death. We explored this model by overexpressing PbpB while strongly repressing PBP-lipo. We discovered overexpressing PbpB completely reversed the growth and morphological defect of PBP-lipo knockdown. This data provided evidence that PBP-lipo and PbpB have overlapping 4,3 cross-linking function at the septum, and that by supplying the cell with more D,D-transpeptidation activity via PbpB, the deleterious effects of PBP-lipo repression were reversed. Interestingly, given that both PBP-lipo and PbpB are essential in *Mab*, the enzymes must have overlapping, but distinct functional roles. Their distinct functions could be revealed by identifying unique genetic or physical interactions each enzyme has with other PG synthesis enzymes. Investigating the unique yet overlapping functions of PbpB and PBP-lipo at the *Mab* septum is an intriguing area of future research.

The two other genes that synergized with PBP-lipo, DacB1 and MAB_0519, are both hypothesized carboxypeptidases that cleave the terminal D-ala-D-ala on the pentapeptide chains of PG monomers forming a tetrapeptide. This substrate is necessary for 3,3 cross-links catalyzed by L-D transpeptidases (*Ghosh et al., 2008*; *Ealand et al., 2019*). Interestingly, DacB1 localizes to the septum in *Mab* and has been described as septally localized in *Msm* and *Mtb* (*Gorla et al., 2018*). In *Mab*, DacB1 also co-localizes with PBP-lipo at the septum. Furthermore, the homolog of MAB_0519 in *Mtb*, Rv3627c, is involved in septal PG synthesis (*Zhang et al., 2019*). Thus, we hypothesize that DacB1 and MAB_0519 are involved in septal PG synthesis and may work with PBP-lipo to ensure proper PG synthesis and remodeling at the septum. Interestingly, DacB1 and MAB_0519 are not functionally redundant, given that both genes are synergistic with PBP-lipo. Further characterization of the PG enzymes in *Mab* and their genetic networks will uncover species-specific nuances in *Mab* PG synthesis that have the potential to be exploited for drug discovery.

Knockdown of PBP-lipo dramatically sensitized the reference *Mab* strain and 11 clinical isolates to several antibiotics. The PBP-lipo knockdown was >128-fold more sensitive to the β-lactams, ampicillin

and amoxicillin, while other β-lactams such as like cephalexin, ticarcillin, and others showed no difference in MIC. This result indicates a specific chemical-genetic interaction between ampicillin/amoxicillin, and PBP-lipo. Historically, β-lactams have not been used to treat *Mab* infections due to high levels of resistance and the presence of β-lactamases (*Soroka et al., 2014*). While imipenem and cefoxitin are used with some success, macrolides and aminoglycosides remain the backbone of *Mab* treatment (*Novosad et al., 2016*). Recently, studies have shown dual β-lactam strategies are effective in killing *Mab* both in vitro and in vivo (*Story-Roller et al., 2019*; *Pandey et al., 2019*; *Lopeman et al., 2020*). These synergy data suggest that understanding the specific antibiotic targets of β-lactams in *Mab* would augment the design of effective dual β-lactam combinations. At the moment, effective combinations are usually tested empirically with no insight into potential mechanisms (*Pandey et al., 2019*). Recently, a potential mechanism for dual-β-lactam synergy in *Mab* was uncovered by Dousa et al. The group discovered that the novel β-lactamase inhibitor, durlobactam, also inhibited a *Mab* carboxypeptidase and by doing so, greatly enhanced *Mab* sensitivity to amoxicillin and imipenem combination therapy (*Dousa et al., 2022*). Another attempt to identify specific targets of PBP targets of β-lactams was conducted by Sayed et al. This group used a combination of a bocillin FL-binding assay to stain all the PBPs in *Mab* followed by proteomics to identify the targets of 12 β-lactams and 2 β-lactamases. Interestingly, in their datasets, PBP-lipo was not identified by mass spectrometry of bocillin FL-bounded PBPs (*Sayed et al., 2020*). This further supports our hypothesis that PBP-lipo is expressed at low levels in *Mab* and is not identifiable by bocillin FL staining under native expression conditions.

Nonetheless, our data demonstrate that using a β-lactam or small molecules that target PBP-lipo would not only kill the cell, but also potentiate β-lactams like ampicillin and amoxicillin. In addition, the permeabilizing effect of PBP-lipo inhibition would potentiate drugs by allowing more compound to enter the cell. Crucially, the observed synergy with PBP-lipo knockdown and β-lactams was not only limited to the reference strain, but also present in 11 clinical isolates of *Mab,* indicating that this synergy can be broadly targeted in *Mab* clinical isolates.

Overall, this work employed TnSeq to identify the uniquely essential gene PBP-lipo in *Mab*. We present PBP-lipo as a promising drug target given both its essentiality in *Mab* cell growth and division and its role in sensitizing *Mab* to a range of antibiotics, including commonly used and accessible antibiotics, ampicillin and amoxicillin. Future studies exploring the PBP networks of mycobacteria can expose further species-specific vulnerabilities in PG synthesis. Finally, identifying chemical inhibitors of PBP-lipo may help form the foundation for new treatments against this emerging and difficult to treat pathogen.

## Materials and methods

A full list of the strains, plasmids, and primers is available in *Supplementary files 7–9* respectively.

### Bacterial strains and culture conditions

All *Mab* and *Mtb* strains were grown shaking at 37°C in liquid 7H9 media consisting of Middlebrook 7H9 salts with 0.2% glycerol, 0.85 g/l NaCl, OADC (oleic acid, 5 g/l albumin, 2 g/l dextrose, 0.003 g/l catalase), and 0.05% Tween80. *Mab* and *Mtb* were plated on Middlebrook 7H10 agar supplemented with 0.5% glycerol. *Msm* was grown shaking at 37°C in liquid on 7H9 media consisting of Middlebrook 7H9 salts with 0.2% glycerol, 0.85 g/l NaCl, ADC, and 0.05% Tween80 and plated on LB agar. Antibiotic selection concentrations for *Mab* cultures were kanamycin 50 µg/µl and zeocin 100 µg/µl. For transposon mutant selection, 100 µg/ml of kanamycin was used. Antibiotic selection concentrations for *Msm* and *Mtb* cultures were kanamycin 25 µg/µl, zeocin 20 µg/µl, and gentamicin 5 µg/µl. For *E. coli* (TOP10, XL1-Blue, and DH5α), all antibiotic concentrations were twice those of the concentrations used for *Msm* and *Mtb*. Induction of all CRISPRi plasmids was performed with 500 ng/µl ATc.

### Strain construction

#### Construction of 0, 1, 2, 3 sgRNA PBP-lipo knockdown strains and PBP-lipo complement strains

To build the 0 sgRNA (empty sgRNA) *Mab* strain, we transformed the pCT296 vector containing an empty sgRNA into *Mab*. To build the 1, 2, and 3 sgRNA strains, first we built individual CRISPRi

plasmids containing single sgRNAs that targeted PBP-lipo. These plasmids were cr25, cr26, and cr56. We then designed primers that amplified the sgRNA targeting region with the appropriate flanking sequences and performed Golden Gate cloning to combine the inserts into one vector. This method was previously described by *Rock et al., 2017*. For the 1 sgRNA strain, we transformed cr25 into the *Mab* strain. For the 2 sgRNA, we transformed the vector pCRC1, which contained the cr25 and cr26 sgRNAs on one backbone. Finally, for the 3 sgRNA strain, we transformed pCRC2, which contained the cr25, cr26, and cr56 sgRNAs on one backbone.

To build the PBP-lipo complement strain, we designed primers to amplify PBP-lipo with modified PAM sequences that were no longer homologous to the 1 and 2 sgRNAs. This recoded PBP-lipo was then cloned into a constitutive expression vector resulting in the pCA323 construct, which was transformed into the 2 sgRNA strain. The catalytically dead mutant version of the recoded PBP-lipo was generated by using primers to generate the S364A mutation. This engineered PBP-lipo was cloned into a constitutive expression vector and the resulting construct, pCA324, was transformed into the 2 sgRNA strain.

### *Msm* and *Mtb* ΔPBP-lipo strains

To generate PBP-lipo knockout mutants in *Msm* and *Mtb,* we used a recombineering approach to replace the endogenous copy of the gene with a zeocin resistance cassette as previously described (*Boutte et al., 2016*). First, 500 base pairs of upstream and downstream sequence surrounding PBP-lipo (*Rv2864c* and *MSMEG_2584c*) were amplified by PCR KOD XtremeTM Hot Start DNA polymerase (EMD Millipore, Billerica, MA). These flanking regions were amplified with overlaps for the zeocin resistance cassette. The two flanking regions and zeocin resistance cassette were then assembled via isothermal assembly (*Gibson et al., 2009*) into a single construct, which was then amplified by PCR. Each deletion construct was transformed into either *Mtb* or *Msm* expressing inducible copies of RecET (*Murphy et al., 2015*). The resulting colonies were screened for the PBP-lipo deletion by PCR.

### PBP-lipo$_{Mab}$-strep expression strain

PBP-lipo$_{Mab}$ was amplified using primers that added a C-terminal strep tag to the protein. The resulting insert was Gibson stitched to the vector backbone that drove expression of PBP-lipo$_{Mab}$-strep from the constitutively active MOP promoter.

### mRFP-PBP-lipo$_{Mab}$

We amplified PBP-lipo$_{Mab}$ in two fragments. The first fragment contained the N-terminal signal sequence of PBP-lipo. The second fragment contained the remainder of the protein. We then amplified mRFP with overhangs that were compatible with the two fragments of PBP-lipo. The resulting three fragments were Gibson assembled to a vector backbone that constitutively drove expression of the mRFP-PBP-lipo$_{Mab}$ fusion protein. A gly-gly-ser-gly-ser-gly linker was used in between mRFP and the two fragments of PBP-lipo.

### DacB1-GFP-strep

DacB1 lacking a stop codon was amplified along with GFP containing overlaps with DacB1 and a gly-gly-ser linker. The two fragments were Gibson assembled to a vector backbone that constitutively drove expression of the DacB1-GFP-strep fusion protein. ftsZ-mNeonGreen and DacB1-mNeonGreen ftsZ and DacB1 were amplified along with mNeonGreen containing a gly-gly-ser linker. Each gene and its corresponding overlapping mNeonGreen fragment were Gibson assembled to a vector that constitutively drove expression from the weak promoter iMyc.

### natP-GFP-ftsZ

To express FtsZ off its natural promoter, 300 bp of sequence upstream of the FtsZ start codon was amplified and Gibson assembled with two additional fragments corresponding to GFP with a gly-gly-ser linker and FtsZ. The resulting vector was an N-terminal GFP-FtsZ fusion expressed off the FtsZ natural promoter.

## Growth curve

*Mab, Msm,* and *Mtb* cultures were grown to log phase and diluted to $OD_{600}$ 0.05. Growth curves were performed in a 96-well format at 37°C with shaking. $OD_{600}$ was measured every 15 min for 24–72 hr using the TECAN plate reader.

## Colony forming units

*Mab* and *Msm* cultures were grown to mid-log phase and then serially diluted in their respective media in 96-well plates. Dilutions were then spot plated onto solid media +/-500 ng/μl ATc. Plates were incubated at 37°C for 4 days (*Mab*) or 3 days (*Msm*). After incubation, the resulting CFUs were calculated by counting the number of colonies at the appropriate dilution. The fraction survivability was calculated as CFUs+ATc/-ATc.

## Minimum inhibitory concentration determination

*Mab, Msm,* and *Mtb* were grown to mid-log phase and diluted to an $OD_{600}$=0.003 in each well of non-treated 96-well plates (Genesee Scientific) containing 100 μl of antibiotic serially diluted in 7H9+OADC + 5 μg/ml clavulanate (Sigma-Aldrich). For MICs on *Mab* knockdown cells, cultures were induced for knockdown 18 hr prior with 500 ng/μl ATc. Wells with knockdown bacteria also contained 500 ng/μl ATc. *Msm* media contained ADC rather than OADC. Cells were incubated with drug at 37°C with shaking for 1 day (*Mab, Msm*) or 7 days (*Mtb*). Afterward, 0.002% resazurin diluted in $ddH_2O$ (Sigma-Aldrich) was added to each well. Plates were then incubated for 24 hr. MICs were determined by the concentration of antibiotic that turned wells blue signifying no metabolic activity (*Kieser et al., 2015b*).

## Generation of transposon libraries

To create transposon mutant libraries of *Mab* ATCC 19977 and *Msm* (wildtype and ΔPBP-lipo), 100 ml of bacterial cultures were grown to an $OD_{600}$ of 1.5–2.0. For *Mab*, cultures were grown in biological triplicate. For *Msm*, the wildtype strain was grown in duplicate while one replicate of the ΔPBP-lipo strain was grown. The cells were then washed twice with 50 ml of MP Buffer (50 mM Tris-HCl pH 7.5, 150 mM NaCl, 10 mM $MgSO_4$, 2 mM $CaCl_2$) and resuspended in 1/10th the culture volume in MP Buffer. Afterward, $2 \times 10^{11}$ pfu of temperature-sensitive φMycoMarT7 phage carrying the Himar1 transposon were added to bacteria. Phage and bacterial cultures were incubated at 37°C for 4 hr with shaking. The transduced cultures were spun down at RT 4000 rpm for 10 min and resuspended in 12 ml of PBS-Tween. The cultures were then tittered by plating on 7H10 plates (*Mab*) or LB (*Msm*) supplemented with kanamycin 100 μg/μml (*Mab*) or 20 μg/μl (*Msm*) before freezing the aliquots for future plating. After determining the titer, 150,000 bacterial mutants of each transduced culture were plated onto kanamycin selective plates and grown for 4 days at 37°C. The resulting mutant libraries were harvested and stored in aliquots with 7H9+10% glycerol at –80°C.

## **Genomic DNA extraction**

To prep libraries for sequencing, we isolated the gDNA of the mutant libraries using a bead-beating protocol developed in the lab. Two to six ml of transposon mutant libraries were thawed and spun down to remove excess glycerol in the sample. Pellets were then resuspended in TE Buffer (10 mM Tris-HCl pH 7.4, 1 mM EDTA pH 8) and placed in bead-beating tubes with 600 μl of 25:24:1 phenol:chloroform:isoamyl alcohol. The samples were then bead-beat 4× for 45 s at 4000 rpm. Samples were cooled on ice for 45 s between each successive bead-beat round. After bead-beating, the samples were spun down for 10 min at 13,000 rpm. The supernatant was transferred to a new falcon tube and incubated with 1:1 volume phenol:chloroform for 1 hr rocking. Samples were then spun down in MaXtract High Density phase-lock tubes (Qiagen) at 1500 *g* for 5 min. This separated the aqueous and organic layers. Afterward, ½ volume of chloroform was added to the tube and the contents were spun down at 1500 *g* for 5 min. The aqueous phase was transferred to a new tube and RNAse A (Thermo Fisher) was added to a final concentration of 25 μg/μl. The tubes were incubated at 37°C for 1 hr with shaking. Finally, the supernatant was washed with 1:1 volume of phenol-chloroform followed by ½ volume of chloroform. The aqueous phase was transferred to a new tube and genomic DNA was precipitated with 1 volume of isopropanol alcohol and 1/10th volume of 3 M sodium acetate pH 5.2.

The library gDNA was washed twice with fresh 70% ethanol and resuspended in nuclease-free ddH$_2$O. This protocol consistently generated high yields of *Mab* gDNA.

## TnSeq, mapping, and analysis

Sequencing libraries were prepared from the extracted gDNA by amplifying chromosomal-transposon junctions following the protocol as described by *Long et al., 2015*. These amplicons were then sequenced using the Illumina Hi-Seq platform. The resulting reads were mapped onto the *Mab* or *Msm* genomes. The data were analyzed using the TRANSIT pipeline (*DeJesus et al., 2015*). Insertion counts at each TA site were normalized using TTR and averaged across replicates. Essentiality categories of genes were determined by the HMM algorithm in TRANSIT. Resampling analysis was used to compare insertion accounts between genes in the wildtype and ΔPBP-lipo *Msm* strains.

## Cloning of CRISPRi constructs

To clone sgRNAs onto the CRISPRi vector backbone, complimentary targeting oligos were annealed and ligated (T4 DNA ligase, NEB) into the BsmBI digested CRISPRi vector backbone. To clone multiple sgRNAs into the same vector, the promoter, sgRNA handle, and sgRNA sequences were amplified with the appropriate flanking sequences to perform Golden Gate cloning using the SapI cloning site in the CRISPRi vector (*Rock et al., 2017*). All cloning was performed using DH5α cells and sequences were verified before transformation into *Mab* and *Msm*.

## Transformation of *Mab*

To generate electrocompetent *Mab* cells, 50 ml of *Mab* cultures were grown to mid-log phase. Cells were spun down at 4000 rpm for 10 min and washed twice with pre-chilled 10% glycerol. Cell pellets were resuspended in 1/100th of the starting volume and kept on ice for immediate use or stored at –80°C for future use. For transformations, ~100 ng of plasmid DNA was incubated with electrocompetent cells for 5 min before electroporation using the settings: 2500 V, 125 Ω, 25 μF. Afterward, cells were recovered in 1 ml of 7H10 with OADC without selection for 3 hr at 37°C with shaking. Cells were then plated on 7H10 with the appropriate selection marker and incubated for 4 days at 37°C.

## Microscopy and image analysis

All imaging was performed on an inverted Nikon TI-E microscope at 60× and 100× magnification. For PBP-lipo knockdown experiments, cultures were induced for knockdown with 500 ng/μl ATc for 18 hr prior to imaging. All *Mab* cultures were fixed with 7H9+3% paraformaldehyde (PFA) for 1 hr and resuspended in PBS + 0.05% Tween80 prior to imaging. Cellular features including cell length, width, and fluorescence signal were analyzed using the MOMIA and GEMATRIA image analysis pipelines developed in the lab (*Zhu et al., 2021*).

## Fluorescent D-amino acid staining

NADA was synthesized by Tocris following the published protocol (*Kuru et al., 2015*). To stain *Mab* with NADA, 0.1 mM of FDAA was added to 1 ml of exponentially growing cells and incubated for 3 min before washing in warm 7H9 twice. For *Mab* imaging, after the second wash, cells were fixed in 7H9+3% PFA for 1 hr. The PFA was then washed off and cells were resuspended in PBS + 0.05% Tween80 prior to imaging.

## RNA extraction and RT-qPCR

For each strain, cultures were grown in biological triplicate to mid-log phase and then diluted back in +/-500 ng/ml ATc and grown for 18 hr to achieve target knockdown. Afterward, 2 OD$_{600}$ equivalents of cells from each culture were harvested by centrifugation, resuspended in TRIzol (Thermo Fisher), and lysed by bead beating (Lysing Matrix B, MP Biomedicals). Total RNA was extracted using the RNA miniprep (Zymo Research). Residual genomic DNA was digested with TURBO DNase (Ambion), and samples were cleaned with RNA clean-up columns (Zymo Research). cDNA was prepared using random hexamers following manual instructions (Life Technologies Superscript IV). Alkaline hydrolysis was then used to remove RNA. cDNA was purified by spin column (Qiagen) and then quantified by RT-qPCR on a Viia7 light cycler (Applied Biosystems) using iTaq Universal SYBR Green Supermix

(BioRad). All qPCR primer pairs were confirmed to be >95% efficient. The masses of cDNA used were experimentally validated to be within the linear dynamic range of the assay. Signals were normalized to the *sigA* (*MAB_3009*) transcript and quantified by the ΔΔCt method. Error bars are standard deviations from three biological replicates.

## Bocillin FL staining

*Mab* cultures were grown to mid-log phase and 10 OD units of bacteria were spun down and resuspended in 1 ml of fresh 7H9 media with 5 µg/ml of clavulanate. Bocillin-BODIPY or FITC were added to cultures to a final concentration of 10 µM. Cells were then incubated at 37°C for 2.5 hr, with aluminum wrapping to prevent light exposure. After labeling, cells are washed with PBS + 0.05% Tween80 three times and resuspended in 200 µl of lysis buffer (50 mM Tris-HCl pH = 7.4, 50 mM NaCl, protease inhibitor [Roche]). Samples were bead beat 4× at 4000 rpm for 1 min with 1 min of rest on ice in between rounds of bead-beats. $CaCl_2$ was added to a final concentration of 1 mM along with 4 U of DNAse I and 10 mg/ml of fresh lysozyme. Samples were incubated at 37°C for 30 min. Afterward, supernatants were normalized by A280 protein concentration, diluted with 6× Laemmli buffer, and run on 4–12% NuPAGE Bis Tris precast gels (Life Technologies). To visualize fluorescence, we used a BioRad Gel Doc system. To detect the presence of PBP-lipo-strep, we performed a Western blot. Membranes were blotted with rabbit α-Strep (GenScript) at 1:1000 in TBST + 3% BSA.

## Calcein staining

For each strain, cultures were grown in biological triplicate to mid-log phase and diluted back in +/-500 ng/ml ATc and grown for 18 hr to achieve target knockdown. Afterward, 3 OD units of bacteria were washed twice with PBS + 0.05% Tween80. Cells were resuspended in PBS + 0.05% Tween80 and 100 µl of cells at $OD_{600}$=0.4 was added to 96-well plates. The plates were incubated in the plate reader for 30 min at 37°C with shaking. Afterward, calcein was added at a final concentration of 1 µg/ml. Fluorescence signal was measured every minute using the Tecan plate reader.

## Lipoprotein extraction

Lipoprotein and non-lipoproteins were extracted using the protocol described by *Armbruster and Meredith, 2018*. Briefly, a *Mab* strain constitutively expressing a strep-tagged PBP-lipo was grown to an OD 1.2. The cells were pelleted and resuspended in Tris-buffered saline/EDTA (TBSE) with 1 mM phenylmethyl sulfonyl fluoride and 0.5 mg/ml lysozyme. Cells were incubated for 20 min at 37°C followed by bead-beating for 8 cycles, 4000 rpm, 45 s per cycle. The cells were then pelleted at 3000× *g* for 5 min at 4°C. TX-114 was added to the resulting supernatant at a final concentration of 2% (vol/vol). The sample was kept on ice and inverted every 15 min for 1 hr. To induce phase separation, the sample was incubated in a 37°C water bath for 10 min followed by centrifugation at 10,000× *g* for 10 min at room temperature. The upper aqueous phase was discarded and ice-cold TBSE was added to the sample to refill the tube to the original volume. The sample was then placed on ice for 10 min and the process of phase separation was repeated for a total of three times. Afterward, 1 volume of ice-cold TBSE was added to the surfactant phase and the sample was centrifuged at 16,000× *g* for 2 min at 4°C. The resulting pellet of insoluble proteins is composed of non-lipoproteins, which was stored for further analysis. The resulting supernatant was transferred to a fresh tube with 1250 µl acetone. The sample was mixed and incubated overnight at –20°C to precipitate the lipoproteins. The next day, the sample was centrifuged at 16,000× *g* for 20 min. The resulting pellet composed of lipoproteins was washed twice with 100% acetone and resuspended in Laemmli SDS-PAGE buffer for further analysis.

# Acknowledgements

This project was funded by the generous support of the Paul and Daisy Soros Foundation, the NIH/NIAID F31 Predoctoral Fellowship award number F31AI149932, and award Number T32GM007753 from the National Institute of General Medical Sciences. The content is solely the responsibility of the authors and does not represent the official views of the National Institute of General Medical Sciences or the National Institutes of Health.

## Additional information

### Funding

| Funder | Grant reference number | Author |
|---|---|---|
| National Institute of Allergy and Infectious Diseases | NIH/NIAID F31AI149932 | Chidiebere Akusobi |
| National Institute of Allergy and Infectious Diseases | R21AI156772 | Bouchra S Benghomari<br>Junhao Zhu<br>Ian D Wolf<br>Shreya Singhvi<br>Charles L Dulberger<br>Thomas R Ioerger<br>Eric J Rubin |
| National Institute of General Medical Sciences | T32GM007753 | Chidiebere Akusobi |

The funders had no role in study design, data collection and interpretation, or the decision to submit the work for publication.

### Author contributions

Chidiebere Akusobi, Conceptualization, Data curation, Formal analysis, Investigation, Methodology, Project administration, Resources, Supervision, Validation, Visualization, Writing - original draft, Writing - review and editing; Bouchra S Benghomari, Conceptualization, Data curation, Formal analysis, Investigation, Methodology; Junhao Zhu, Data curation, Formal analysis, Investigation, Methodology, Software, Validation, Visualization, Writing - review and editing; Ian D Wolf, Data curation, Investigation, Methodology, Validation, Visualization, Writing - review and editing; Shreya Singhvi, Formal analysis, Methodology; Charles L Dulberger, Conceptualization, Data curation, Formal analysis, Investigation, Methodology, Software, Visualization, Writing - review and editing; Thomas R Ioerger, Data curation, Formal analysis, Methodology, Resources, Software, Visualization, Writing - review and editing; Eric J Rubin, Conceptualization, Data curation, Formal analysis, Funding acquisition, Investigation, Project administration, Supervision, Validation, Writing - review and editing

### Author ORCIDs

Chidiebere Akusobi (iD) http://orcid.org/0000-0002-1611-0015
Junhao Zhu (iD) http://orcid.org/0000-0002-3301-5677
Charles L Dulberger (iD) http://orcid.org/0000-0002-1334-5468
Eric J Rubin (iD) http://orcid.org/0000-0001-5120-962X

### Decision letter and Author response

Decision letter https://doi.org/10.7554/eLife.71947.sa1
Author response https://doi.org/10.7554/eLife.71947.sa2

## Additional files

### Supplementary files

• Supplementary file 1. Summary of *Mycobacterium abscessus* subsp. *abscessus* ATCC 19977 transposon sequencing (TnSeq) libraries.

• Supplementary file 2. List of essential genes in *Mycobacterium abscessus,* non-essential genes in *Mycobacterium tuberculosis* (*Mtb*).

• Supplementary file 3. List of essential genes in *Mycobacterium abscessus* (*Mab*) with *non-essential* orthologs in *Mycobacterium tuberculosis* (*Mtb*).

• Supplementary file 4. Transposon sequencing (TnSeq) summary of *Mycobacterium smegmatis* (*Msm*) libraries.

• Supplementary file 5. Minimum inhibitory concentration (MIC) (µg/ml) of wildtype mc²155 and ΔPBP-lipo.

• Supplementary file 6. Minimum inhibitory concentration (MIC) (µg/ml) of wildtype H37Rv and ΔPBP-lipo.

- Supplementary file 7. Bacterial strains used.
- Supplementary file 8. Plasmids used.
- Supplementary file 9. Primers used.
- Transparent reporting form

## Data availability

Transposon sequencing data for the Mycobacterium abscessus strains have been deposited here: https://github.com/ioerger/TTN-Fitness/tree/main/demodata/abscessus/ATCC_19977.

The following dataset was generated:

| Author(s) | Year | Dataset title | Dataset URL | Database and Identifier |
|---|---|---|---|---|
| Akusobi C, Benghomari BS, Ioerger TR, Rubin EJ | 2021 | TTN-Fitness | https://github.com/ioerger/TTN-Fitness/tree/main/demodata/abscessus/ATCC_19977 | GitHub, GitHub |

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
