## [Editor Report]

This study reports the results of a transposon inactivation screen to identify essential genes in Mycobacterium abscessus. The authors investigate one hit, the gene encoding a putative class B penicillin-binding protein, PBP-lipo. They confirm that the PBP-lipo gene is essential despite the presence of a homologous gene and that PBP-lipo is present in other mycobacteria, but not essential in these. They further characterize the consequences of PBP-lipo gene depletion in M. abscessus and demonstrate that the gene product is required for maintaining cell morphology, whilst also participating in a network with other cell wall enzymes.

---

## [Decision Letter]

**Decision letter after peer review:**

Thank you for submitting your article "High-density transposon mutagenesis in *Mycobacterium abscessus* identifies an essential penicillin-binding lipo-protein (PBP-lipo) involved in septal peptidoglycan synthesis and antibiotic sensitivity" for consideration by *eLife*. Your article has been reviewed by 3 peer reviewers, and the evaluation has been overseen by Bavesh Kana as the Reviewing and Senior Editor. The reviewers have opted to remain anonymous.

Essential revisions:

1. There is no evidence that PBP-lipo is a lipoprotein although the authors presented this as a certainty (lines 143-144). Several PBPs have a cysteine residue within or near the end of the N-terminal TM region, but these are not lipoproteins. For example, *E. coli* PBP3 has a cysteine at position 30 shortly after the TM region, but it does not appear to be a lipoprotein. The authors should either prove that PBP-lipo is a lipoprotein, which has been achieved for lipoproteins in other species by mass spectrometry and/or labelling with radioactive palmitate. If such evidence cannot be provided, they should rename the enzyme, removing 'lipo' from the name, and refrain from presenting the protein as a lipoprotein.

2. There appears to be a problem with the mRFP-PBP-lipo construct in which mRFP is tagged at the N-terminus of PBP-lipo (line 210). If PBP-lipo is indeed a lipoprotein, as assumed by the authors and perhaps confirmed in the above-mentioned point, then this N-terminal tagging may not be the best approach. Lipoproteins are typically cleaved at the N-terminus of the cysteine residue, which would remove the mRFP tag from the protein and make the cellular localization experiments meaningless. Considering these concerns, the authors should:

a. Prove that the construct is functional and can complement cells depleted of WT PBP-lipo, and

b. Subject cell extracts with Bocillin labelling and confirm the presence of an mRFP-PBP-lipo band or

c. Detect mRFP-PBP-lipo in cell extracts with an antibody against mRFP.

(A is essential and either b or c would be acceptable)

Also, in line 211 they should state whether mRFP-PBP-lipo was expressed in the presence (or not) of WT PBP-lipo.

3. Reviewers appreciated the experiments to investigate the PG networks in Mab and other mycobacteria which revealed, for example, the genetic interactions between the PBP-lipo gene and the dacB1, pbpB and MAB_0519 genes. However, to support the appending conclusions, the authors need to confirm that DacB1 and PBPB were depleted by the CRISPR technology to similar extent in Mab and Msm, to exclude the possibility that differences in genetic interactions were due to different efficiency in the CRISPR gene depletion in the two species. This is particularly important for Msm, as there were no genetic interactions (lines 270-2760). Hence, they should perform Bocillin assays and quantify the depletion of DacB1 and PBPB in both species.

4. Control experiments to show that the DacB1-GFPmut3 fusion is functional should be added.

5. To strengthen their model, the authors should co-localize PBPlipo with another divisome component.

6. The authors should Colocalise mRFP-PBP-lipo (if functional) with DacB1-GFP (if functional).

7. It is known that several bacteria have homologous PBPs that cannot fully replace each other and have specific roles under different growth conditions. The authors don't present or discuss such examples in their Discussion. The discussion should be expanded beyond mycobacteria to include examples from other species, such as:

– *Salmonella* has a dedicated class B PBP3 homologue (called PBP3-SAL) next to its 'normal' PBP3 required for cell division. PBP3-SAL is specifically required for cell division in acidified phagosomes (mBio. 2017 Dec 19;8(6):e01685-17. doi: 10.1128/mBio.01685-17).

– *E. coli* uses a specialised DD-CPase PBP6B to maintain cell morphology when growing under acidic conditions (mBio. 2016 Jun 21;7(3):e00819-16. doi: 10.1128/mBio.00819-16.).

Such examples can use used to discuss whether PBP-lipo and its homologue might not be essential under certain growth conditions at which the homologue is highly active.

*Reviewer #1 (Recommendations for the authors):*

This manuscript was a pleasure to read and easy to evaluate. It is interesting and exceptionally well done.

Two questions that occurred to me are: (1) is the localisation of PBP-lipo affected in the presence of AMX or AMP, and( 2) is it possible to find a β-lactam that would sensitize WT Mab to AMX or AMP to the same degree as the PBP-lipo knock-down Mab strains.

The title refers to PBP-lipo as a lipoprotein, but no such evidence that it is indeed a lipoprotein has been provided.

*Reviewer #2 (Recommendations for the authors):*

I would like to ask the authors to address the following points:

1. There is no evidence that PBP-lipo is a lipoprotein although they presented this as a certainty (lines 143-144). To my knowledge, several PBPs have a cysteine residue within or near the end of the N-terminal TM region, but these are not lipoproteins. For example, *E. coli* PBP3 has a cysteine at position 30 shortly after the TM region, but it does not appear to be a lipoprotein. They should prove that PBP-lipo is a lipoprotein, which has been achieved for lipoproteins in other species by mass spectrometry and/or labelling with radioactive palmitate. If they cannot provide such evidence, they should rename the enzyme, removing 'lipo' from the name, and refrain from presenting the PBP as a lipoprotein.

2. It is known that several bacteria have homologous PBPs that cannot fully replace each other and have specific roles under different growth conditions. Unfortunately, the authors don't present or discuss such examples in their Discussion. I suggest that they expand their discussion beyond mycobacteria to include examples from other species, for example:

– *Salmonella* has a dedicated class B PBP3 homologue (called PBP3-SAL) next to its 'normal' PBP3 required for cell division. PBP3-SAL is specifically required for cell division in acidified phagosomes (mBio. 2017 Dec 19;8(6):e01685-17. doi: 10.1128/mBio.01685-17).

– *E. coli* uses a specialised DD-CPase PBP6B to maintain cell morphology when growing under acidic conditions (mBio. 2016 Jun 21;7(3):e00819-16. doi: 10.1128/mBio.00819-16.).

They could include such examples and discuss whether PBP-lipo and its homologue might not be essential under certain growth conditions at which the homologue is highly active? In my view they cannot exclude such a possibility, as they did not assay growth phenotypes at different growth conditions.

3. I see a main problem with the mRFP-PBP-lipo construct in which mRFP is tagged to the N-terminus of PBP-lipo (line 210). If PBP-lipo is indeed a lipoprotein, as assumed by the authors and perhaps confirmed in additional experiments (my point 1 above), then this N-terminal tagging cannot work. Lipoproteins are typically cleaved at the N-terminus of the cysteine residue, which would remove the mRFP tag from the protein and make the cellular localization experiments meaningless. Because of these doubts regarding the mRFP-PBP-lipo construct they should:

– prove that the construct is functional and can complement cells depleted of WT PBP-lipo,

– subject cell extracts with Bocillin labelling and confirm the presence of an mRFP-PBP-lipo band and

– detect mRFP-PBP-lipo in cell extracts with an antibody against mRFP.

Also, in line 211 they should state whether mRFP-PBP-lipo was expressed in the presence (or not) of WT PBP-lipo.

4. I appreciate the experiments to investigate the PG networks in Mab and other mycobacteria which revealed, for example, the genetic interactions between the PBP-lipo gene and the dacB1, pbpB and MAB_0519 genes. However, in my view they should confirm that DacB1 and PBPB were depleted by the CRISPR technology to similar extent in Mab and Msm, to exclude the possibility that differences in genetic interactions were due to different efficiency in the CRISPR gene depletion in the two species. This is particularly important for Msm, as there were no genetic interactions (lines 270-2760). Hence, they should perform Bocillin assays and quantify the depletion of DacB1 and PBPB in both species.

5. Figure 10B. They should add control experiments to show that the DacB1-GFPmut3 fusion is functional (Bocillin labelling of extracts; Western Blot with anti-Gfp).

*Reviewer #3 (Recommendations for the authors):*

The experiments largely support the conclusion derived. However, I feel the following experiments/clarifications will further strengthen some of the conclusions in the manuscript:

(i) Colocalization of PBP-lipo with at least one of the divisome components may strengthen the argument that PBP-lipo localizes to the division septa.

(ii) Similarly the idea that PBP-lipo interacts with PbpB and/or DacB1 can be further strengthened by at least colocalising mRFP-PBP-lipo with DacB1-GFP.

(iii) Is the localization of PBP-lipo affected in PbpB or DacB1 or MAB_0519 depleted cells? Are the localization of these proteins interdependent?

(iv) Does depletion of pbpB or dacB1 or MAB_0519 make Mab cells susceptible to antibiotics as during PBP-lipo knockdown?

[Editors’ note: further revisions were suggested prior to acceptance, as described below.]

Thank you for resubmitting your work entitled "High-density transposon mutagenesis in *Mycobacterium abscessus* identifies an essential penicillin-binding lipo-protein (PBP-lipo) involved in septal peptidoglycan synthesis and antibiotic sensitivity" for further consideration by *eLife*. Your revised article has been evaluated by Bavesh Kana (Senior Editor) and a Reviewing Editor.

The manuscript has been improved but there is one major concern that needs to be addressed, as outlined below.

Reviewers remain concerns about the experiments supporting the annotation of PBP-Lipo as a lipoprotein:

1. Please provide sufficient detail on the lipoprotein extraction procedure. The information is not available in Ref 26, which also appears to focus on Vibrio, rather than Mycobacteria.

2. Results of the attempts to verify that PBP-lipo has a lipid modification are shown in Figure 3 —figure supplement 3B. However, this figure shows that the majority of PBP-lipo (>90%?) seems to segregate into the non-LP fraction. Quantification is not possible because the non-LP lane is overloaded but the results suggest the absence of a lipoprotein modification. The Methods section lacks sufficient information about this experiment (how were the LP and non-LP fractions obtained?), and there is no positive or negative control of a verified lipoprotein or non-lipoprotein. As a result, reviewers indicate that this experiment does not address the issue of the Lipo modification. You should either rename the enzyme, removing "lipo" from the name, or state very clearly in the manuscript that the lipo modification is hypothetical.

Other concerns:

L. 273: insert "genetic" in front of "PG network" in the heading.

L. 309: format problem.

Figure 6 —figure supplement 5A. the y-axis should read "log (or ln?) fold change mRNA expression" instead of "fold change mRNA expression".

L507: PbpB is 4-3 crosslinking enzyme. It does not necessarily follow that PBP-lipo must have the same activity. The ectopic complementation could be due to other crosslinks that also stabilize PG. Please reconsider this statement.

---

## [Author Response]

Essential revisions:1. There is no evidence that PBP-lipo is a lipoprotein although the authors presented this as a certainty (lines 143-144). Several PBPs have a cysteine residue within or near the end of the N-terminal TM region, but these are not lipoproteins. For example, *E. coli* PBP3 has a cysteine at position 30 shortly after the TM region, but it does not appear to be a lipoprotein. The authors should either prove that PBP-lipo is a lipoprotein, which has been achieved for lipoproteins in other species by mass spectrometry and/or labelling with radioactive palmitate. If such evidence cannot be provided, they should rename the enzyme, removing 'lipo' from the name, and refrain from presenting the protein as a lipoprotein.

PBP-lipo has previously been identified computationally as a lipoprotein though, as the reviewers point out, these predictions are not always reliable. To determine if it truly is acylated, we performed an extraction and found that PBP-lipo segregates into the lipoprotein fraction (Figure 3 —figure supplement 3).

2. There appears to be a problem with the mRFP-PBP-lipo construct in which mRFP is tagged at the N-terminus of PBP-lipo (line 210). If PBP-lipo is indeed a lipoprotein, as assumed by the authors and perhaps confirmed in the above-mentioned point, then this N-terminal tagging may not be the best approach. Lipoproteins are typically cleaved at the N-terminus of the cysteine residue, which would remove the mRFP tag from the protein and make the cellular localization experiments meaningless. Considering these concerns, the authors should:a. Prove that the construct is functional and can complement cells depleted of WT PBP-lipo, andb. Subject cell extracts with Bocillin labelling and confirm the presence of an mRFP-PBP-lipo band orc. Detect mRFP-PBP-lipo in cell extracts with an antibody against mRFP.(A is essential and either b or c would be acceptable)Also, in line 211 they should state whether mRFP-PBP-lipo was expressed in the presence (or not) of WT PBP-lipo.

Created Figure 5 —figure supplement 1 to address these revisions.

– S Figure 5 —figure supplement 1A shows a schematic of the mRFP-PBP-lipo fusion protein. mRFP was inserted after the Cysteine residue and attached to linkers on the 5’ and 3’ region. The site of the single sgRNA binding site that was mutated to produce a recoded version of the construct is depicted as well

– Figure 5 —figure supplement 1B depicts a fluorescent Western Blot against 2 strains, one expressing mRFP-PBP-lipo and the other expressing PBP-lipo. Both constructs were strep-tagged. The anti-strep Western Blot shows the mRFP-PBP-lipo fusion protein runs at the expected molecular weight if both proteins were fused

– Figure 5 —figure supplement 1C is a growth curve showing that both the recoded and non-recoded versions of mRFP-PBP-lipo complements the growth defect caused by the native PBP-lipo knockdown. The recoded construct fully complements the phenotype while the non-recoded version partially complements

Added text to explain that the mRFP-PBP-lipo fusions are expressed in the context of native PBP-lipo expression that is either unrepressed or repressed with ATc. Also we explicitly mentioned in lines 227-228 that the fusion protein is expressed in the presence of the native PBP-lipo protein

3. Reviewers appreciated the experiments to investigate the PG networks in Mab and other mycobacteria which revealed, for example, the genetic interactions between the PBP-lipo gene and the dacB1, pbpB and MAB_0519 genes. However, to support the appending conclusions, the authors need to confirm that DacB1 and PBPB were depleted by the CRISPR technology to similar extent in Mab and Msm, to exclude the possibility that differences in genetic interactions were due to different efficiency in the CRISPR gene depletion in the two species. This is particularly important for Msm, as there were no genetic interactions (lines 270-2760). Hence, they should perform Bocillin assays and quantify the depletion of DacB1 and PBPB in both species.

We performed qPCR to measure level of repression of *pbpB*, *dacB1,* and *MAB_0519* in both the Mab double knockdown and *Msm* PBP-lipo knockout strains (Figure 6 —figure supplement 5A). The level of knockdown in both species for all three genes were comparable and not statistically significantly different from each other. Since the CRISPR system we use acts at the level of transcriptional repression, this provides better quantitation than protein methods such as Bocillin labeling.

4. Control experiments to show that the DacB1-GFPmut3 fusion is functional should be added

We performed a Western Blot on the DacB1-GFPmut3-strep protein and detected the presence of a band ~70kDa, which is the predicted size of the fusion protein (Figure 6 —figure supplement 4A). Since DacB1 is redundant and non-essential, there is no real assay for function so there is no way to prove that it is functional.

5. To strengthen their model, the authors should co-localize PBPlipo with another divisome component

We performed a co-localization experiment with mRFP-PBP-lipo and FtsZ-mNeonGreen (mNG). Figure 5A shows co-localization of PBP-lipo and FtsZ in representative cells. In total, we analyzed a total of 100+ cells for each strain. We then used software developed in the lab (Junhao et al., 2022) to plot the septal fluorescence signal as a function of cell length for both PBP-lipo and FtsZ (Figure 6B).

This experiment showed that both FtsZ and PBP-lipo localize to the septum, with FtsZ localizing to the septum first. This result has been published in several bacteria. Finally, we showed that knockdown of PBP-lipo leads to the disruption of FtsZ localization, with cells forming multiple FtsZ rings spread throughout the length of the cell

6. The authors should Colocalise mRFP-PBP-lipo (if functional) with DacB1-GFP (if functional).

Figure 6 —figure supplement 4C shows co-localization of mRFP-PBP-lipo and DacB1-mNeonGreen Due to the toxicity of co-expressing dual tagged PBPs with fluorescent proteins, DacB1-mNeonGreen was expressed of a weak promoter to allow for the co-localization experiments to be performed

7. It is known that several bacteria have homologous PBPs that cannot fully replace each other and have specific roles under different growth conditions. The authors don't present or discuss such examples in their Discussion. The discussion should be expanded beyond mycobacteria to include examples from other species, such as:– Salmonella has a dedicated class B PBP3 homologue (called PBP3-SAL) next to its 'normal' PBP3 required for cell division. PBP3-SAL is specifically required for cell division in acidified phagosomes (mBio. 2017 Dec 19;8(6):e01685-17. doi: 10.1128/mBio.01685-17).– *E. coli* uses a specialised DD-CPase PBP6B to maintain cell morphology when growing under acidic conditions (mBio. 2016 Jun 21;7(3):e00819-16. doi: 10.1128/mBio.00819-16.).Such examples can use used todiscuss whether PBP-lipo and its homologue might not be essential under certain growth conditions at which the homologue is highly active.

Thank you. We have added these points to the Discussion section.

[Editors’ note: further revisions were suggested prior to acceptance, as described below.]

Reviewers remain concerns about the experiments supporting the annotation of PBP-Lipo as a lipoprotein:1. Please provide sufficient detail on the lipoprotein extraction procedure. The information is not available in Ref 26, which also appears to focus on Vibrio, rather than Mycobacteria.

The reference that details the lipoprotein extraction procedure is Ref 29, Armbruster and Meredith, 2018 (L193). The protocol details steps to generally extract bacterial lipoprotein without any specific focus on a bacterial species. A similar lipoprotein protocol extraction protocol was applied to mycobacteria (see reference 30 which is referred to below).

Included a brief description of the protocol was described in the Results section (L194-196) and a full description of the protocol was added in the Materials and methods (L787-805).

2. Results of the attempts to verify that PBP-lipo has a lipid modification are shown in Figure 3 —figure supplement 3B. However, this figure shows that the majority of PBP-lipo (>90%?) seems to segregate into the non-LP fraction. Quantification is not possible because the non-LP lane is overloaded but the results suggest the absence of a lipoprotein modification. The Methods section lacks sufficient information about this experiment (how were the LP and non-LP fractions obtained?), and there is no positive or negative control of a verified lipoprotein or non-lipoprotein. As a result, reviewers indicate that this experiment does not address the issue of the Lipo modification. You should either rename the enzyme, removing "lipo" from the name, or state very clearly in the manuscript that the lipo modification is hypothetical.

– Added ‘lipoprotein extraction’ section in the Methods section that details how the lipoprotein and non-lipoprotein fractions were obtained.

– Included a reference by Young and Garbe (Ref 30) where lipoproteins extracted from *Mycobacterium tuberculosis* were present both in the lipoprotein and non-lipoprotein fractions

– Given the valid critiques of the experiment, we have stated clearly in the manuscript, both in the abstract and during the first mentions of *MAB_3167c* in the introduction and Results sections, that the gene encodes a ‘penicillin binding protein and hypothetical lipoprotein’ (PBP-lipo). Furthermore, we added language stating that future experiments are needed to confirm if the protein is indeed lipidated.

– Edited the title of the manuscript by removing PBP-lipo. Title now reads “High-density transposon mutagenesis in *Mycobacterium abscessus* identifies an essential penicillin-binding protein involved in septal peptidoglycan synthesis and antibiotic sensitivity”

Other concerns:L. 273: insert "genetic" in front of "PG network" in the heading.

Change made

L. 309: format problem.

Problem fixed

Figure 6 —figure supplement 5A. the y-axis should read "log (or ln?) fold change mRNA expression" instead of "fold change mRNA expression".

These changes were not plotted on a log scale, and so we have kept the y-axis title as ‘fold change mRNA expression.’ The same y-axis legend is used in Figure 3 —figure supplement 1B

L507: PbpB is 4-3 crosslinking enzyme. It does not necessarily follow that PBP-lipo must have the same activity. The ectopic complementation could be due to other crosslinks that also stabilize PG. Please reconsider this statement.

Changed language to “investigating the unique yet overlapping functions of PbpB and PBP-lipo at the *Mab* septum is an intriguing area of future research.” (L521-522)